



Natural Hazards
and Earth System
Sciences

# Non-stationary analysis of water level extremes in Latvian waters, Baltic Sea, during 1961–2018

**Nadezhda Kudryavtseva[1], Tarmo Soomere[1,2], and Rain Männikus[1]**

[1]Wave Engineering Laboratory, Department of Cybernetics, School of Science, Tallinn University of Technology, Akadeemia 21, Tallinn, 12618, Estonia
[2]Estonian Academy of Sciences, Kohtu 6, Tallinn, 10130, Estonia

**Correspondence:** Nadezhda Kudryavtseva (nadia@ioc.ee)

**Abstract.** Analysis and prediction of water level extremes in the eastern Baltic Sea are difficult tasks because of the contribution of various drivers to the water level, the presence of outliers in time series, and possibly non-stationarity of the extremes. Non-stationary modeling of extremes was performed to the block maxima of water level derived from the time series at six locations in the Gulf of Riga and one location in the Baltic proper, Baltic Sea, during 1961–2018. Several parameters of the generalized-extreme-value (GEV) distribution of the measured water level maxima both in the Baltic proper and in the interior of the Gulf of Riga exhibit statistically significant changes over these years. The most considerable changes occur to the shape parameter $\xi$. All stations in the interior of the Gulf of Riga experienced a regime shift: a drastic abrupt drop in the shape parameter from $\xi \approx 0.03 \pm 0.02$ to $\xi \approx -0.36 \pm 0.04$ around 1986 followed by an increase of a similar magnitude around 1990. This means a sudden switch from a Fréchet distribution to a three-parameter Weibull distribution and back. The period of an abrupt shift (1986–1990) in the shape parameters of GEV distribution in the interior of the Gulf of Riga coincides with the significant weakening of correlation between the water level extremes and the North Atlantic Oscillation (NAO). The water level extremes at Kolka at the entrance to the Gulf of Riga reveal a significant linear trend in shape parameter following the $\xi \approx -0.44 + 0.01(t - 1961)$ relation. There is evidence of a different course of the water level extremes in the Baltic proper and the interior of the Gulf of Riga. The described changes may lead to greatly different projections for long-term behavior of water level extremes and their return periods based on data from different intervals.

**Highlights.**

– Water level extremes in the eastern Baltic Sea and the Gulf of Riga are analyzed for 1961–2018.

– Significant changes in parameters of generalized-extreme-value distribution are identified.

– Significant linear trend in shape parameter is established at Kolka.

– The shape parameter changes in a step-like manner.

– The shape parameter of GEV has regime shifts around 1986 and 1990 in the gulf.

## 1 Introduction

Extreme values are the most common input for coastal design and management (Coles, 2004). Observed or measured time series of water level usually serve as the most reliable source of information. However, a sophisticated approach to a problem (extent of a flood, height of a structure, and so forth) requires not only the values of extremes but also their frequency (e.g., return periods of different heights) and the duration of extreme events. As time series of observed water level are commonly not longer than 100 years, there have been attempts to find suitable theoretical statistical distributions of extreme values which could be used to find reliable values for return periods. This is a complicated issue

since the data may be too short, inaccurate, or non-stationary (Mudersbach and Jensen, 2010; Galiatsatou et al., 2019). Moreover, there could be different populations of storms which result in extreme values which do not follow the chosen distribution (Suursaar and Sooäär, 2007).

The situation is even more complicated in estuarine-type environments, where a multitude of drivers may contribute to the formation of high water levels (Del-Rosal-Salido et al., 2019). In the Baltic Sea, the frequent presence of long-term aperiodic high water levels in the entire sea (Lehmann and Post, 2015; Lehmann et al., 2017) may contribute to storm surges depending on the location, openness, and orientation of single coastal sections. For example, in the eastern regions of the Baltic Sea, the largest storm surges are caused by strong westerly winds that often also push large volumes of water into this sea (Lehmann et al., 2017).

Depending on the method in use, set of data, and regional differences in the storm surge heights, the estimates of water level extremes commonly reveal the disparity between different models and observations (Bardet et al., 2011). This feature was thoroughly analyzed in Meier et al. (2004) using two different circulation models and two sea level scenarios. Dieterich et al. (2019) demonstrated that the estimates of water level extremes for several areas of the Baltic Sea such as the Skagerrak, Gulf of Finland, and Gulf of Riga are sensitive to the choice of the particular regional climate (circulation) model even if forced by the same external drivers. The uncertainties in projections of extreme water levels can be made smaller by an increase in the model resolution (Kowalewski and Kowalewska-Kalkowska, 2017). This approach inter alia makes it possible to resolve the nonlinear response of the water level extremes to the increase in the mean water level in shallow regions (Gräwe and Burchard, 2012). Different drivers of extreme water levels interact in a nonlinear manner so that their joint impact may be more significant than the sum of effects of single drivers (Hieronymus et al., 2017; Kudryavtseva et al., 2020). The most complicated dynamics seem to occur in the eastern sub-basins of the Baltic Sea, where conventional methods for extreme-value estimation are not able to accommodate all observed and hindcast extremes (Suursaar and Sooäär, 2007). Moreover, the spread among different methods can be substantial in areas that may have extensive wave setup (Eelsalu et al., 2014).

The large-scale atmospheric teleconnections characterizing the North Atlantic, such as the North Atlantic Oscillation (NAO) and Arctic Oscillation (AO), exhibit a well-known correlation with the mean sea level of the Baltic Sea (e.g., Andersson, 2002; Lehmann et al., 2002; Dailidienė et al., 2006; Suursaar and Sooäär 2007) and its eastern sub-basins (Männikus et al., 2020). Specifically, some 88 % of water level variations in the Baltic Sea can be explained by the pattern of atmospheric pressure over the Baltic Sea (Karabil et al., 2018). This correlation is the highest in winter (DJFM) and is most likely caused by a strong impact of the wind conditions over the North Sea on the Baltic Sea mean sea level.

The positive phases of NAO are characterized by stronger westerly winds and more frequent storms, which push the water from the North Sea to the Baltic Sea through the Danish straits.

The correlation between the sea level and NAO exhibits a remarkable variability with time, which became stronger in the 20th century (e.g., Andersson, 2002; Jevrejeva et al., 2005). One possible explanation is drifting of the actual Icelandic low-pressure center with time, which is not visible in the NAO index time series since it is measured between two fixed locations (e.g., Andersson, 2002). The drift of this center will result in a change in the regional wind properties and their correlation with the fixed NAO index. It is also possible that an interplay of several large-scale atmospheric teleconnections is driving the sea level variability in the Baltic Sea region.

The relation between the extreme sea levels and the teleconnections, however, is not widely studied in the Baltic Sea region, especially in terms of its time variability. Jaagus and Suursaar (2013) show a positive correlation between the NAO and AO indices and water level monthly maxima along the Estonian coast and a negative correlation with the Scandinavia index (SCAND). The East Atlantic, East Atlantic–Western Russia, and Polar indices, on the other hand, do not show any significant correlations. Marcos and Woodworth (2017) demonstrate that the 99th percentile of the sea level is related to the NAO index along the Swedish and Finnish coasts, and it is independent of the mean sea level variations. Männikus et al. (2020) discuss a significant correlation between the NAO annual index and the annual sea level maxima, minima, and standard deviations observed in the Gulf of Riga. The correlation is different during the seasons and is most pronounced in winter (DJFM).

The analysis of Weisse et al. (2014) signals an increase in the Baltic Sea water level extremes in the past 100 years. The main contributor to this process is the increase in the mean sea level. This is consistent with the outcome of the analysis of the water volume extremes of the Baltic Sea over 2 centuries (Ekman, 1996). An increase in wind speed will lead to a stronger water level reaction in areas such as the Gulf of Finland, where extremes are already high (Hieronymus et al., 2018). A specific feature of the Baltic Sea is that extreme water levels may increase faster than the mean water level even without an increase in the wind speed (Meier, 2006; Soomere and Pindsoo, 2016). This property seems to be distinctive to the eastern sub-basins such as the Gulf of Finland, the West Estonian archipelago and the Gulf of Riga (Fig. 1), and the southeastern segments of the sea (Pindsoo and Soomere, 2020).

A natural reflection of this difference in the increase in water level extremes is the extensive spatial variation in the parameters of the generalized-extreme-value (GEV) distribution for water level extremes along the northeastern Baltic Sea shore (Soomere et al., 2018). This variation includes alongshore changes in the sign of the GEV shape parame-

ter. It signals the necessity of using different particular cases (three-parameter Weibull, Gumbel, or Fréchet) of distribution for adequate projections of extreme water levels and their return periods in different coastal segments. The situation is even more complicated in changing climates where the background process of formation of high water levels is not necessarily statistically stationary, and the parameters of the GEV distribution of extreme water levels may change in time (Kudryavtseva et al., 2018). Such variations in these parameters may lead to great variations in projections of the resulting water level for more extended return periods.

Spatio-temporal variations in the parameters of the GEV distribution in the Baltic Sea basin have been so far analyzed based on modeled water levels (e.g., Kudryavtseva et al., 2018; Soomere et al., 2018). Most of the existing studies into extreme water levels in the eastern Baltic Sea have been performed under the assumption of stationarity of the underlying extreme-value distributions. Only Suursaar and Sooäär (2007) address possible changes in the parameters of the GEV distribution for different periods in Pärnu for 1923–2005. A comparison of projections of extreme water levels and their return periods for modeled and measured data indicates that local effects may substantially contribute to the extreme water levels in specific locations (Eelsalu et al., 2014). This outcome motivates more detailed research into long-term data sets of water level. The availability of high-quality long-term measured-water-level data sets from the shore of the central Baltic Sea region makes it possible to extend this analysis to in situ data.

In this paper, we focus on the temporal behavior of the parameters of distributions of extreme water levels in Latvian waters (Fig. 1). The most interesting and least studied from the viewpoint of the water level extremes part of the study area is the Gulf of Riga. Extreme water levels in this sub-basin are, historically, the third-highest in the entire Baltic Sea after the eastern Gulf of Finland and southwestern Baltic Sea (Dailidienė et al., 2006; Averkiev and Klevannyy, 2010). This feature reflects a specific pattern of the formation of water levels in this semi-enclosed water body (Astok et al., 1999; Männikus et al., 2019). For completeness, we include a comparison with the properties of extreme water levels at one tide gauge (Pärnu) in Estonian waters.

The main objective of this study is to characterize the temporal course and quantify the magnitude of temporal changes to the parameters of the GEV at selected observation sites. We start from a description of the study area, data, and methods for the analysis of extreme water levels in Sect. 2. The analysis is based on block maxima of water levels over the windy autumn and winter season. Differently from annual maxima, such maxima are not serially correlated. The nature of changes to various parameters of the GEV is analyzed in Sect. 3. We also test different approximations which could describe the patterns of change in GEV parameters and roughly estimate the role of non-stationarity of the data in the

formation of the values of parameters of GEV. The discussion and conclusions are presented in Sect. 4.

## 2 Data and methods

### 2.1 Observed data in the study area

The study area – the shores of Latvia with a total length of about 500 km – consists of two major segments. About half of the study area on the western coast of Latvia is open to the Baltic proper (Fig. 1). The water level regime and the behavior of water level maxima in this segment, represented by the Liepāja and Kolka tide gauges, are mostly similar to the relevant features in Lithuania (Dailidienė et al., 2006) and the western shores of the West Estonian archipelago (Suursaar and Sooäär, 2007; Eelsalu et al., 2014; Soomere et al., 2018). Another half of the study area is located on the western, southern, and eastern shores of the Gulf of Riga (Fig. 1). The tide gauge at Pärnu is located about 70 km north of the border between Latvia and Estonia. This gulf has a generally regular size with dimensions of about $130 \times 140$ km (Suursaar et al., 2002) and is connected with the Baltic proper via relatively narrow and shallow (systems of) straits. The primary connection, Irbe Strait, is 27 km wide, but the water depth in the sill area is mostly < 10 m (Maritime Administration of Latvia, 2014). The connections of another outlet via the Väinameri (Moonsund) sub-basin in the West Estonian archipelago sea are much narrower (the width of, e.g., Suur Strait is 4–5 km) and shallower, with a sill depth of about 5 m.

Irbe Strait is open towards the Baltic proper to the southwest, that is, to one of the predominant wind directions (Soomere, 2003). This configuration of the Gulf of Riga supports a two-step mechanism of formation of extreme water levels (Astok et al., 1999). As mentioned above, specific atmospheric forcing patterns may drive a large volume of water into the Baltic Sea on weekly scales (Post and Kõuts, 2014; Lehmann et al., 2017). Such massive volume changes may increase the average sea level in the entire Baltic Sea so much that the sea level in the eastern Baltic Sea may persist by 60–80 cm over the long-term mean for many weeks (Soomere and Pindsoo, 2016). A similar process may additionally increase the average sea level in the entire Gulf of Riga so that water level in its eastern and southern parts exceeds the sea level at the Baltic-proper shores of Latvia by another 1 m (Männikus et al., 2019). Even though such highly elevated water levels usually persist only a few days in this gulf, they are usually driven by strong westerly winds over Irbe Strait. Therefore, they are often associated with high local storm surges in the western parts of the gulf.

Extremely high water levels in the Gulf of Riga are thus developed under the joint impact of three major drivers: water volume of the entire Baltic Sea with a characteristic timescale of weeks, water occasionally pushed by a sequence

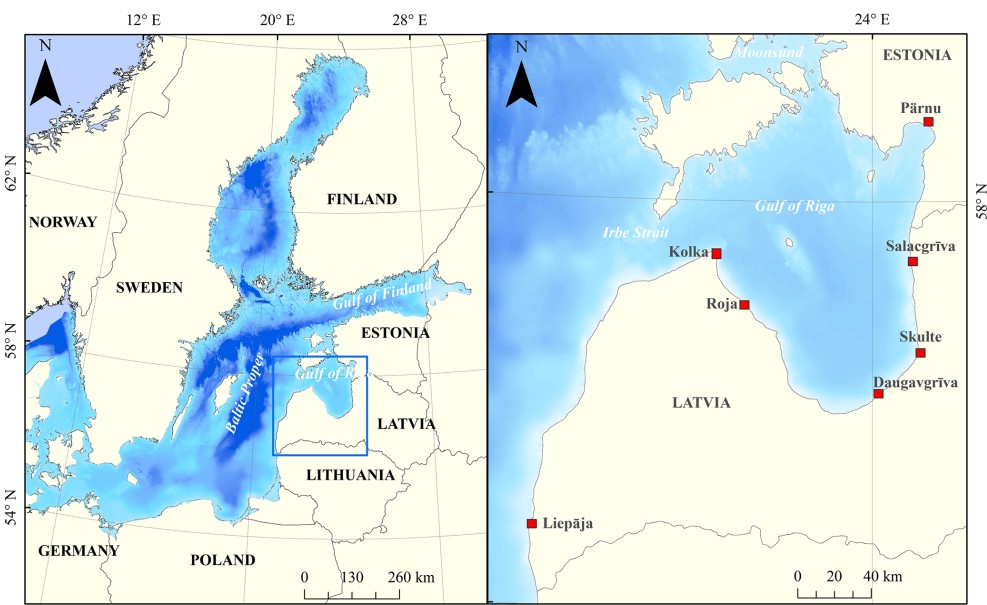

**Figure 1.** Analyzed water level measurement sites (red rectangles) in the Gulf of Riga and on the Baltic-proper shores of Latvia.

of cyclones into the Gulf of Riga for 1–2 d (Suursaar et al., 2002), and local storm surges with a typical duration of a few hours. Each of these drivers may add about 1 m to the resulting water level (Männikus et al., 2019). The joint impact of these processes has likely led to extreme water levels from 2.47 m at Skulte to 2.75 m at Pärnu (Averkiev and Klevannyy, 2010).

Water level observations have been carried out at various locations on the Latvian coast over almost 2 centuries. Currently, the Latvian Environment, Geology and Meteorology Centre (LEGMC) operates tide gauges at 10 sites. The readings of observations and measurements from 1961 are available on their website (https://www.meteo.lv/hidrologija-datu-pieejamiba/). The sampling frequency, coverage, and completeness of recordings vary greatly between the locations. Hourly records started mostly in the middle of the 2000s, when automatic devices were installed. The properties and quality of these time series are presented in Männikus et al. (2020). In this study, time series from seven stations (Liepāja, Kolka, Roja, Daugavgrīva, Skulte, and Salacgrīva in Latvian waters and Pärnu in Estonian waters; Fig. 1, Table 1) are used as these are the most reliable in terms of monthly completeness. The Estonian Weather Service provided the data set for Pärnu.

From 1 December 2014 the height system LAS200.5 (European Vertical Reference System, EVRS) is used in Latvia. Before that, the official height system BK77, with reference level associated with the Kronstadt zero, was used. This zero was defined as the average water level at this location in 1825–1840 (Lazarenko, 1986). As the information about water levels in Latvia and neighboring Baltic countries published in the international literature until 2019 is given in the

BK77 system, we shall use water level data in this system as well. Moreover, other authors (Averkiev and Klevannyy, 2010) have also given results in the BK77.

The quality of data in these stations is analyzed in detail in Männikus et al. (2019, 2020). The interplay of the water masses in the lake of Liepāja and the canal connecting the lake with the sea may on some occasions greatly affect the readings of extreme water levels. Seasonal course of water level at Daugavgrīva is influenced by the hydroelectric plant about 20 km upstream of the river; however, this impact does not affect annual maxima of water levels at this site.

The water level data from Pärnu tide gauge before 2005 have been analyzed in Suursaar and Sooäär (2007). The shapes of empirical probability distributions of the occurrence of different water levels at Liepāja, Daugavgrīva, and Pärnu (Fig. 2) were evaluated in Männikus et al. (2019). The completeness of the data set of hourly observations is $\geq 99.54\%$. The recordings at Liepāja represent water levels at the eastern shore of the Baltic proper. The site at Kolka is also strongly influenced by the water level in the Baltic proper. However, the Daugavgrīva and Pärnu data characterize water level in the southern and northeastern bayhead of the Gulf of Riga, respectively.

Both empirical probability distributions have a quasi-Gaussian appearance, which is characteristic in the northeastern Baltic Sea (Johansson et al., 2001). This shape of the probabilities reflects the joint impact of storm surges (that follow a Poisson distribution on open ocean shores; Schmitt et al., 2018) and frequently existing large volumes of excess water. The excess water is pumped into the Baltic Sea by specific sequences of atmospheric processes (Leppäranta and Myrberg, 2009) and exhibits a Gaussian distribu-

**Nat. Hazards Earth Syst. Sci., 21, 1–18, 2021**           **https://doi.org/10.5194/nhess-21-1-2021**

**Table 1.** The main parameters of the observation locations, basic properties of water level (presented in the BK77 system), and hourly data completeness for 1 January 1961–31 December 2018 in these locations. The maximum water level is extracted from the official data's current version (Männikus et al., 2020). The data set for Pärnu from 1 January 1961 till 31 December 2018 was provided by the Estonian Weather Service. The slashes indicate different values and their occurrence time in different sources. The year in brackets in the second column indicates the beginning of systematic measurements that have been discontinued.

| Location | Measurements since | Coordinates | Mean level (cm) | Maximum level (cm) with date | Minimum level (cm) with date | Hourly data completeness 1961–2018 |
|---|---|---|---|---|---|---|
| Liepāja | 1 Jan 1865 | 56°30′56″ N, 20°59′58″ E | 2.0 | 174 (18 Oct 1967) | −86 (18 Jan 1972) | 99.69 % |
| Kolka | 1 Jan 1884 | 57°44′13″ N, 22°35′34″ E | 1.2 | 161/134 (9 Jan 2005/ 18 Oct 1967) | −113 (3 Nov 2000) | 35.25 % |
| Roja | (1932) 01 Nov 1949 | 57°30′24″ N, 22°48′06″ E | −1.0 | 167/160 (9 Jan 2005/ 18 Oct 1967) | −89 (28 Jan 2010) | 30.28 % |
| Daugavgrīva | 1 Jan 1875 | 57°3′34″ N, 24°1′24″ E | 9.2 | 224 (2 Nov 1969) | −107 (14 Oct 1976) | 99.98 % |
| Skulte | 1 Jan 1939 | 57°18′57″ N, 24°24′34″ E | 6.1 | 231/247 (2 Nov 1969) | −109 (14 Oct 1976) | 93.65 % |
| Salacgrīva | 1 Oct 1928 | 57°45′19″ N, 24° 21′13″ E | 5.8 | 215 (28 Mar 1968) | −116 (14 Oct 1976) | 29.16 % |
| Pärnu | (1893) 1 Nov 1949 | 58°22′55″ N, 24°28′38″ E | 5.2 | 275 (9 Jan 2005) | −121 (14 Oct 1976) | 99.54 % |

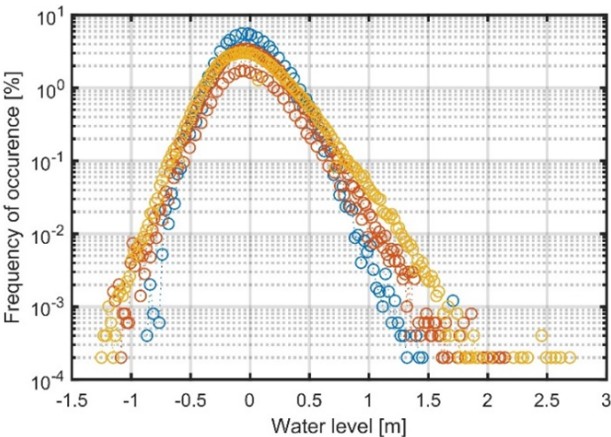

**Figure 2.** Empirical distributions of the frequency of occurrence of different water levels at Liepāja (blue), Daugavgrīva (red), and Pärnu (yellow). Adapted from Männikus et al. (2019).

tion (Soomere et al., 2015a). The skewed shapes of the distributions for Liepāja and Daugavgrīva (skewness 1.431 and 1.674, respectively) indicate a well-known property of the eastern Baltic Sea that elevated water levels are more likely than negative surges (Johansson et al., 2001; Suursaar and Sooäär, 2007; Männikus et al., 2019).

## 2.2 Projections based on block maxima

Long-term water level time series can be analyzed and interpreted using the extreme-value theory results if certain fundamental conditions are fulfilled. The essential requirements are that the selected maximum or minimum values in the time series to be used, e.g., for the block maximum method (Coles, 2004), must be (i) statistically independent and (ii) identically distributed. In other words, each value should be an independent, random sample from the same population.

To achieve statistical independence and remove possible serial correlation, one should consider only values that are sufficiently separated in time. Monthly water level maxima are often correlated because there is a significant time lag between the impact of large-scale atmospheric patterns and the reaction of water level (Johansson et al., 2014). As the events of elevated water level of the entire Baltic Sea (Post and Kõuts, 2014; Samuelsson and Stigebrandt, 1996) may last a few months (Soomere and Pindsoo, 2016), often annual maximum or minimum values at every observation station are considered in the analysis of extremes (e.g., Méndez et al., 2007). However, sometimes annual extremes may be correlated as well. The same cluster of storms may be reflected by two subsequent annual maxima (one in December, another in January of the next year).

Männikus et al. (2020) highlighted the well-known seasonal variability in water level course in the eastern Baltic Sea. It was shown that the most reliable way to select uncorrelated water level extremes is to use the maxima of the entire relatively windy season starting from late spring or early summer (e.g., in July) and ending in June of the next year. This extended windy season is called the stormy season for brevity. The set of water level maxima for stormy seasons is suitable for analyzing extreme water levels and their return periods in the eastern Baltic Sea (Soomere and Pindsoo, 2016).

The difference between the two sets of block maxima (annual and stormy season) is generally insignificant. However, substantial differences are seen in the projections of extreme water levels and their return periods. Eelsalu et al. (2014) reported about 10 cm differences for the observed data in Esto-

nian coastal waters, whereas high water levels projected using the maxima of stormy seasons were usually higher than those based on the annual maxima. For the above-listed reasons, we employed block maxima for stormy seasons defined as time periods from 1 July until 30 June of the subsequent year.

## 2.3 Extreme-value distributions

The water level data from tide gauges listed in Table 1 were used "as is" as presented on the LEGMC website. To avoid the impact of nonzero average water level on the outcome of the analysis of extremes of water level (Coles, 2004), the time series were de-meaned. The water level maxima for each stormy season were found directly from the de-meaned time series. Projections of extreme water levels and their return periods were constructed based on the theoretical generalized-extreme-value distribution (Coles, 2004). The use of this distribution is justified if samples are independent and identically distributed random variables. The Gumbel and Weibull distributions are, in fact, particular cases of a GEV distribution:

$$G_{st}(x; \mu, \sigma, \xi) = \exp\left\{-\left[1 + \xi\left(\frac{x - \mu}{\sigma}\right)\right]^{-1/\xi}\right\}. \qquad (1)$$

The GEV distribution is characterized by a location parameter $\mu \in R$, scale parameter $\sigma \in R$, and shape parameter $\xi \in R$ (Coles, 2004). Here $x$ has the meaning of block maxima (e.g., maximum water level for each stormy season). The return period $T(\hat{x})$ for a particular value $x$ is given by the $1/[1 - T(\hat{x})]$th percentile of $G(x)$

$$T(\hat{x}) = \frac{1}{1 - G(\hat{x})}. \qquad (2)$$

For the shape parameter $\xi \to 0$, the GEV distribution reduces to a Gumbel distribution. If $\xi < 0$, the GEV distribution is equivalent to a three-parameter Weibull distribution and for $\xi > 0$ to a Fréchet distribution (Fig. 3f). A Gumbel distribution is best suitable for the description of extremes of populations described by distributions with an exponential tail such as the Gaussian distribution. A Weibull distribution matches extremes of populations with so-called light-tailed (very rapidly decaying) distributions. The Fréchet distributions have a strong positive tail.

The shapes of empirical water level distributions at Liepāja and Daugavgrīva resemble a Gaussian distribution but are skewed towards high water levels (the skewness is 1.431 and 1.674, respectively). Their kurtosis (3.3 and 4.2 for Liepāja and Daugavgrīva, respectively) is slightly different from the value of kurtosis of a Gaussian distribution (Eq. 3). Therefore, the probability of very high water level values differs from their expectation for a Gaussian-distributed data set.

A typical feature of the northeastern Baltic Sea coastal waters is the presence of outliers in the water level recordings that do not follow the classic extreme-value distributions

(Fig. 4). The threshold for defining the outliers could be evaluated from the difference between the first and third quartile of the sample (Suursaar and Sooäär, 2007). The difference is multiplied by 1.5 and added to the third quartile to reach the threshold. These outliers could be created, for example, by a sequence of storms that first raised water level considerably in the entire Baltic Sea and then created a "usual" surge in single locations against the background of strongly elevated offshore water level. In this sense, these outliers may represent a different population of water level extremes. On some occasions, the outliers could be explained by local effects, for example, by substantial river discharge or even by reading or measurement error (cf. Männikus et al., 2019). For example, the highest water level at Liepāja (174 cm) was recorded on 18 October 1967 at 14:00 UTC+3 during an event when the water level rose from 60 to 174 cm within 2 h, remained constant for 5 h, and then dropped to 100 cm in 1 h. The water level was likely very high this day at Liepāja; however, it is unlikely that the water level was constant for 5 h.

The presence of a few outliers typically insignificantly impacts the integral parameters of the entire distribution but may have an impact on the parameters of the associated extreme-value distributions and projections of return periods of very high water levels (Suursaar and Sooäär, 2007). For example, using a Gumbel distribution would eventually underestimate the importance of positive outliers and lead to underestimation of values of higher return periods. The opposite bias can be expected from the use of a Weibull distribution. Hence, it would be reasonable to consider various distributions for long-term projections.

To evaluate the parameters of the GEV (including three-parameter Weibull, Fréchet, and Gumbel distributions and the two-parameter Weibull distribution), we used a built-in procedure of maximum-likelihood estimation "*gevfit*" and "*wblfit*" in MATLAB. As it is not possible to decide beforehand which theoretical distribution at best describes water levels and their extremes on the Latvian coast, we also employed other methods for calculating the parameters of these extreme-value distributions. We used a method of moments, which resulted in unbiased and biased estimates of parameters. Finally, we used a statistical module of Hydrognomon, a freely available general-purpose software tool, to evaluate the parameters of the GEV and Gumbel distributions (http://hydrognomon.org/, last access: 8 March 2021). It is an open-source application (http://hydrognomon.org/) running on standard Microsoft Windows platforms and also part of the https://openmeteo.org/ (last access: 9 March 2021) framework. This software employs typical hydrological applications, such as homogeneity tests, stage–discharge analysis, areal integration of point data, hydrometric data processing, evapotranspiration modeling, and lumped hydrological modeling.

## 2.4 Non-stationary-extreme-value analysis

To get an estimate of the level of non-stationarity of the extreme-value distribution, a sliding-window approach was used. The time series was separated into 30-year-long consecutive windows. In each window, a stationary-GEV distribution (Eq. 1) was fitted to the block maxima $x$ with a fixed location parameter $\mu$, scale parameter $\sigma$, and shape parameter $\xi$. Before the fit, the background non-stationarity of the time series caused by the joint impact of global sea level rise and local postglacial uplift is removed by subtracting the annual mean from the block maxima.

If the results of the sliding-GEV test revealed that the parameters of the distribution are time-variable, a non-stationary-GEV analysis described below was performed. In such a case, it is assumed that the parameters are functions of time $\mu = M(t)$, $\sigma = \Sigma(t)$, and $\xi = \Xi(t)$. With this hypothesis, a non-stationary-GEV distribution is fitted with one parameter changing with time (either location, scale, or shape parameter). An illustration of the method is shown in Fig. 3.

For example, to test if the shape parameter is changing with time (Fig. 3f), the following distribution is fitted:

$$G_{\text{nonst}}(x; \mu, \sigma, t) = \exp\left\{ -\left[ 1 + \xi\left( \frac{x - \mu}{\sigma} \right) \right]^{-\frac{1}{\Xi(t)}} \right\}. \quad (3)$$

For the non-stationary-GEV fit, we used the *ismev* package (version 1.42) in the R programming language (version 3.6.1). Various functions describing the time variability in the parameters were applied, including a linear dependence

$$M(t) = \mu_0 + \mu_1 t, \; \Sigma(t) = \sigma_0 + \sigma_1 t, \text{ or } \Xi(t) = \xi_0 + \xi_1 t, \quad (4)$$

quadratic dependence

$$M(t) = \mu_0 + \mu_1 t + \mu_2 t^2, \; \Sigma(t) = \sigma_0 + \sigma_1 t + \sigma_2 t^2,$$
$$\Xi(t) = \xi_0 + \xi_1 t + \xi_2 t^2, \quad (5)$$

and a piecewise constant step function

$$M(t) = \begin{cases} 0, t < t_0 - \Delta t \\ a, t \geq t_0 - \Delta t, \; t \leq t_0 + \Delta t \; . \\ 0, t < t_0 + \Delta t \end{cases} \quad (6)$$

Equation (6) defines a simple model of an abrupt change in the parameter of interest with an amplitude $a$ at time instant $t_0$ and the event duration $\Delta t$. To test whether a particular time dependence of the water level extremes is better than a stationary-GEV fit, a likelihood ratio test is applied.

An increase in the location parameter $\mu$ with time represents the case when the whole GEV distribution is shifted to the right, towards higher values. This means that all extremes, from the most severe ones down to "medium-range" and "low-range" ones, are getting higher (e.g., Kudryavtseva et al., 2018). A decrease in this parameter would lead to

changes in the opposite direction. The scale parameter $\sigma$ is responsible for the width of the distribution. An increase in the scale parameter indicates that "medium-range" extremes are getting more frequent, whereas a decrease corresponds to less frequent "medium-range" extremes.

Changes to the shape parameter $\xi$ may have more complicated consequences. This parameter of a GEV distribution defines the overall type (shape) of the extreme-value distribution. In particular, a change in the shape parameter sign corresponds to a switch between the radically different types of extreme-value distribution. For example, if the negative values of the shape parameter shift towards values close to 0, the extremes that are initially characterized by a Weibull distribution with an upper limit start to follow a Gumbel distribution gradually. Further increase in this parameter means a switch to a Fréchet distribution with a lower limit (Fig. 3). A Gumbel distribution describes best the extremes following distributions with an exponential tail such as the Gaussian distribution. A Weibull distribution matches extremes of populations with very rapidly decaying distributions. The Fréchet distributions are characterized by a strong positive tail.

## 3 Results

### 3.1 Temporal changes in the parameters of the GEV distribution

To estimate the magnitude of changes to the parameters of the GEV at selected observation sites (equivalently, to reveal whether extreme water levels may follow a non-stationary process), a sliding-GEV fit was performed. The time series of stormy-season maxima of water levels (Fig. 3b) for all seven measurement sites were divided into consecutive intervals (windows) with a constant width. A stationary-GEV (Eq. 1) fit was performed for each time interval. The window size was thoroughly tested, and the optimal value of 30 years was selected as it provided the lower noise level.

The location parameter $\mu$ did not show significant changes at most of the measurement sites. A 95 % confidence level was used to define significance. Its values remained almost constant at some locations on the Baltic-proper shore (e.g., Liepāja; Fig. 5a) as well as at some places of the eastern Gulf of Riga such as Daugavgrīva and Salacgrīva. At the other sites (Skulte, Pärnu, Roja, and Kolka), this parameter slightly decreased by about 10 % (Fig. 5). The data from the Skulte tide gauge on the eastern shore of the Gulf of Riga exhibited the most substantial drop in the location parameter, from 104 to 90 cm (Fig. 5).

From these measurement locations, Kolka is primarily affected by the water level in the Baltic proper, Roja is located at the western shore of the Gulf of Riga, and Pärnu is located in the northeastern bayhead of the Gulf of Riga. The 95 % confidence intervals of estimates of this parameter obtained

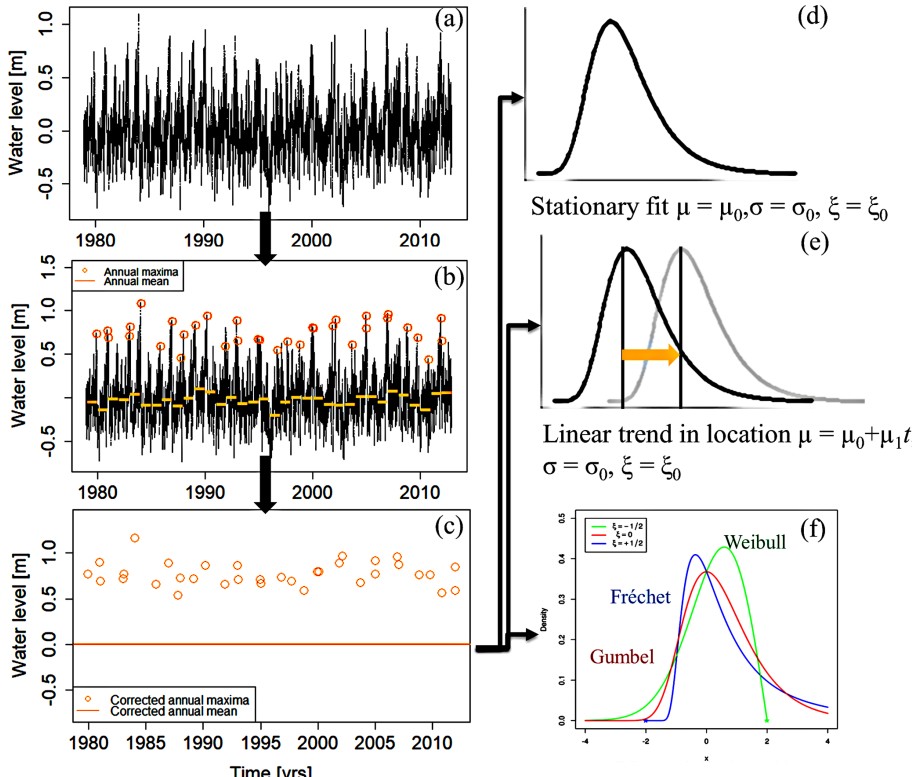

**Figure 3.** The sequence of operations applied to the water level time series: **(a)** the time series is extracted from the model; **(b)** background non-stationarity is removed by subtracting the annual mean from the data; **(c)** a series of maxima of different stormy seasons is created; **(d)** stationary and non-stationary-GEV distributions with changing location **(e)**, scale, or shape **(f)** parameters are fitted to the data.

from the GEV fitting are rather large and overlap for single years. For example, considering the uncertainties in the location parameter at Skulte can be $\sim 98$ (lower level) in 1975 and $\sim 99$ (upper level) in 2001 (Fig. 5). Therefore, it is safe to say that the location parameter of the GEV distribution of water level extremes has not experienced any substantial changes in Latvian waters since the 1960s.

Contrarily, the analysis of the time series using the sliding-GEV fit revealed considerable changes in the scale parameter $\sigma$ at some locations, indicating a spatial variability in its values. The scale parameter experienced a slight decrease of $\sim 20\,\%$ during 1981–1990 at Pärnu (Fig. 6g), Salacgrīva (Fig. 6f), and Skulte (Fig. 6e). An even more substantial increase of 25 % was observed at Liepāja (Fig. 6a). However, other tide gauges did not show significant variability in this parameter.

## 3.2 Regime shifts in terms of the shape parameter

The most dramatic changes of $> 50\,\%$ were observed in the shape parameter $\xi$ (Fig. 7). This feature indicates significant non-stationarity of stormy-season maxima in terms of the shape of the GEV distribution. The temporal course of the shape parameter obtained with the sliding-GEV fit is different at different stations. At Liepāja, the shape parameter was

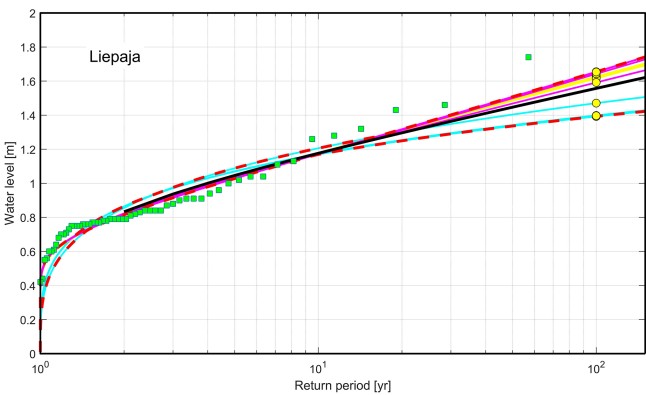

**Figure 4.** Return periods of extreme water levels in Liepāja evaluated using two-parameter Weibull (cyan), Gumbel (yellow), and GEV (magenta) distributions. Green squares indicate block maxima, and yellow circles show different values for a return period of 100 years. Bold dashed red lines show limits of minima and maxima of the ensemble of projections. The bold black line indicates the mean of projections. There are six positive outliers at Liepāja.

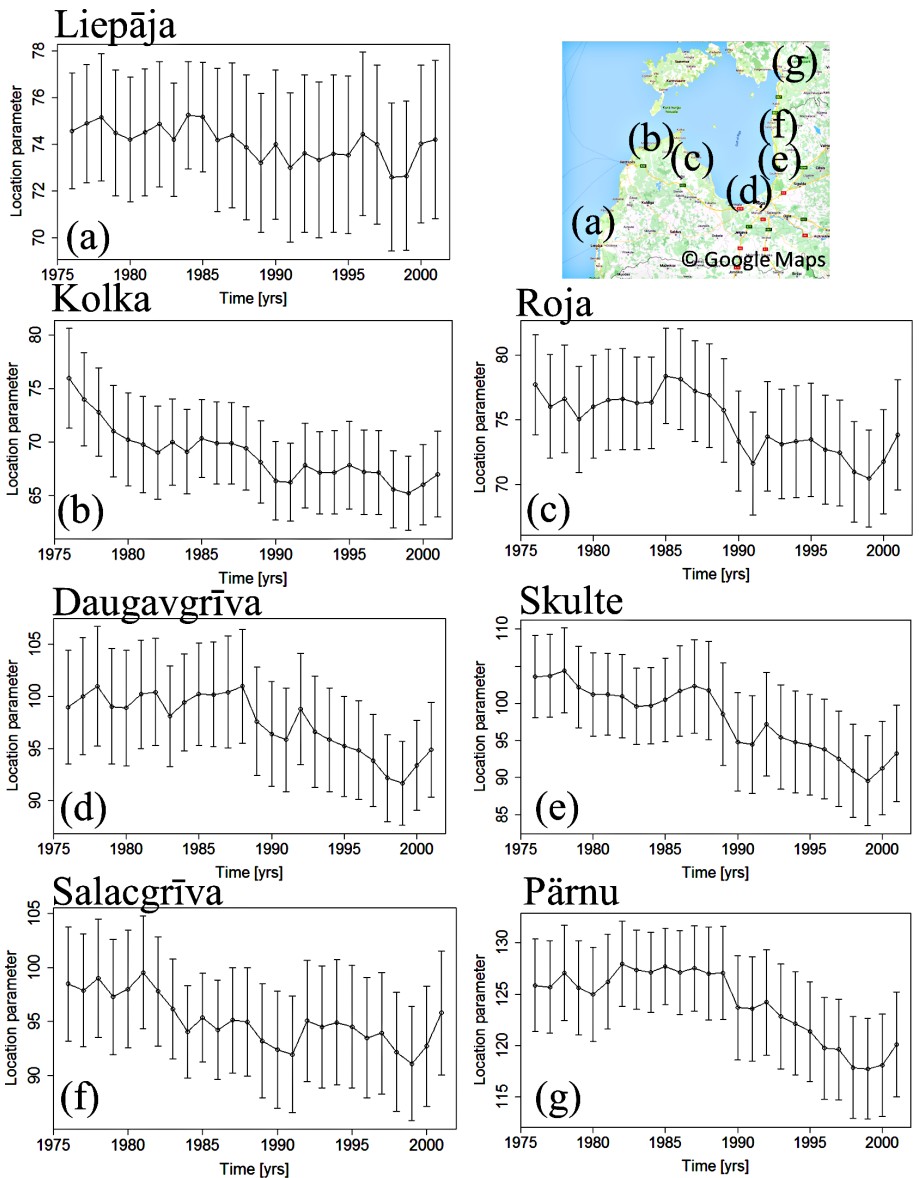

**Figure 5.** Location parameter $\mu$ as a function of time for all tide gauges and 95 % confidence intervals of its estimates. The presented estimates were derived using a sliding-GEV fit with a window of 30 years. The time on the plots corresponds to the time in the middle of this window.

between 0.1 and 0.2 until the mid-1980s and dropped to almost 0 from 1985. In other words, a Fréchet distribution was replaced by a Gumbel one. This parameter was negative (between −0.4 and −0.2) at Kolka until the end of the 1980s and then rapidly increased to about 0.1. Therefore, a Weibull distribution of water level extremes was replaced by a Fréchet distribution at this location.

The values of the shape parameter at all other stations experience a regime shift: a drastic abrupt drop around 1985 followed by an increase of a similar magnitude around 1990. Before and after this drop, the shape parameter was close to 0 at most stations. At Skulte and Salacgrīva it stabilized on

the level of −0.1, whereas it increased to about 0.1 at Daugavgrīva. During the years 1985–1990, it was from about −0.2 (Salacgrīva and Daugavgrīva) down to about −0.4 at Roja, Pärnu, and Skulte (Fig. 7, Table 2).

Importantly, Fig. 7 demonstrates that the features of temporal changes in the shape parameter $\xi$ of the relevant GEV are different at the locations reflecting (or strongly affected by) water levels in the Baltic proper (see above) and in the interior of the Gulf of Riga. The properties of water level at all measurement sites in the interior of the Gulf of Riga show consistent behavior. Before 1985, the shape parameter is consistently close to 0 in the sense that its difference from 0 is

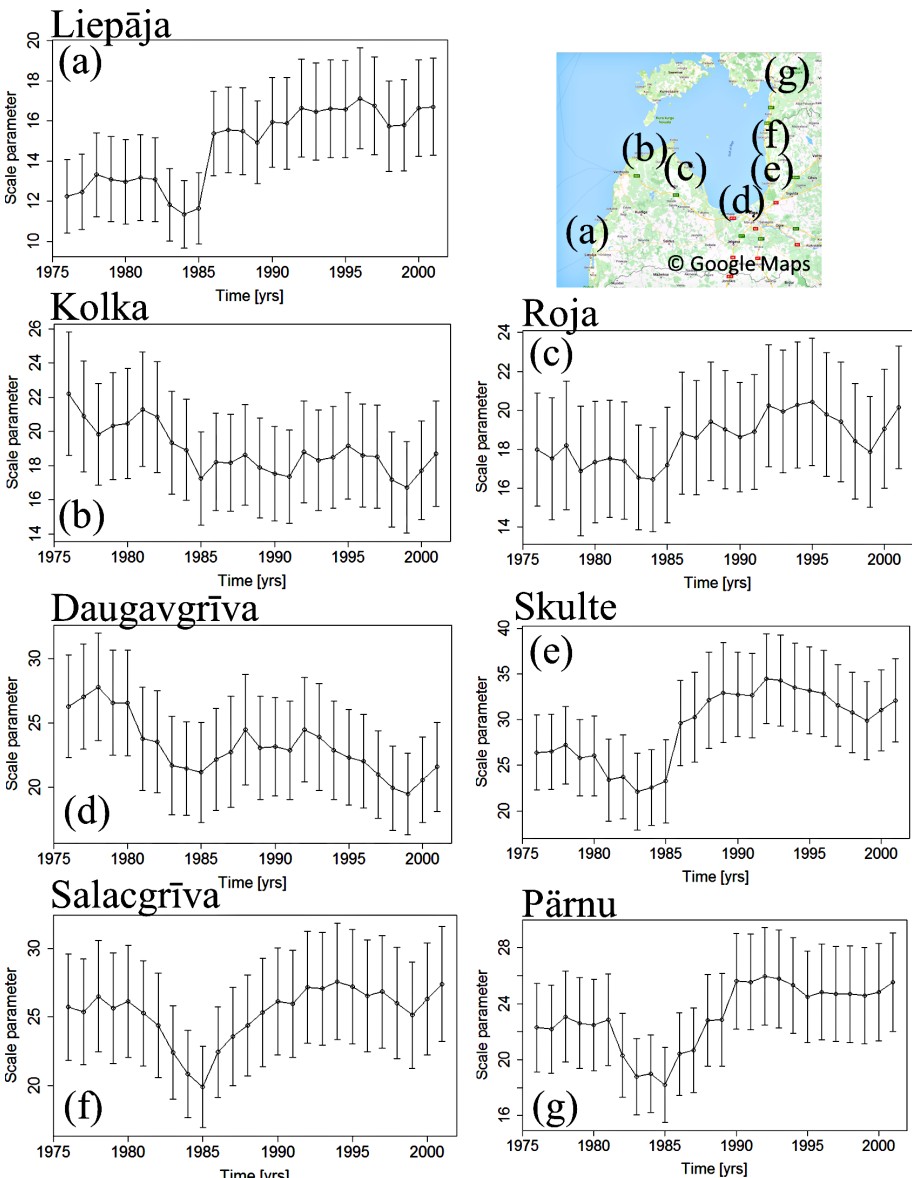

**Figure 6.** Scale parameter $\sigma$ as a function of time for Liepāja (top), Skulte (middle), and Pärnu (bottom) tide gauges. The presented estimates were derived using a sliding-GEV fit with a window of 30 years. The time on the plots corresponds to the time in the middle of this window.

less than the level of uncertainty (95 % confidence intervals) of its evaluation. Thus, differently from the Baltic proper, the GEV distribution follows a Gumbel one in the Gulf of Riga.

During the regime shift in 1985–1989, all stations (except for Daugavgrīva) show consistently negative values of the shape parameter, with an average value over all stations of $\xi \approx -0.36 \pm 0.04$. Therefore, the extreme values of water level followed a three-parameter Weibull distribution. Only the data from Daugavgrīva reveal a considerable uncertainty in the sense that 95 % confidence limits involve the 0 value for most of the years in 1985–1989. Therefore, its switch to negative values is not as clear as for other sites.

After 1989, the recordings show a higher discrepancy in $\xi$. At Daugavgrīva, Pärnu, and Roja $\xi \approx -0.0 \pm 0.02$ within the uncertainties, and thus a Gumbel distribution is acceptable again. The estimates for the shape parameter of the GEV distribution for Salacgrīva and Skulte return to a negative (but smaller) value $\xi \approx -0.12 \pm 0.04$.

The test for non-stationarity employs a sliding window of 30 years. For example, the GEV parameters listed for the year 1985 are evaluated using the extreme-water-level data from 1970 to 2000. Therefore, the abrupt changes in the shape parameter did not necessarily occur specifically in 1985 or 1989. As the reported values of the parameters characterize the GEV fit of extreme water levels over 30 years,

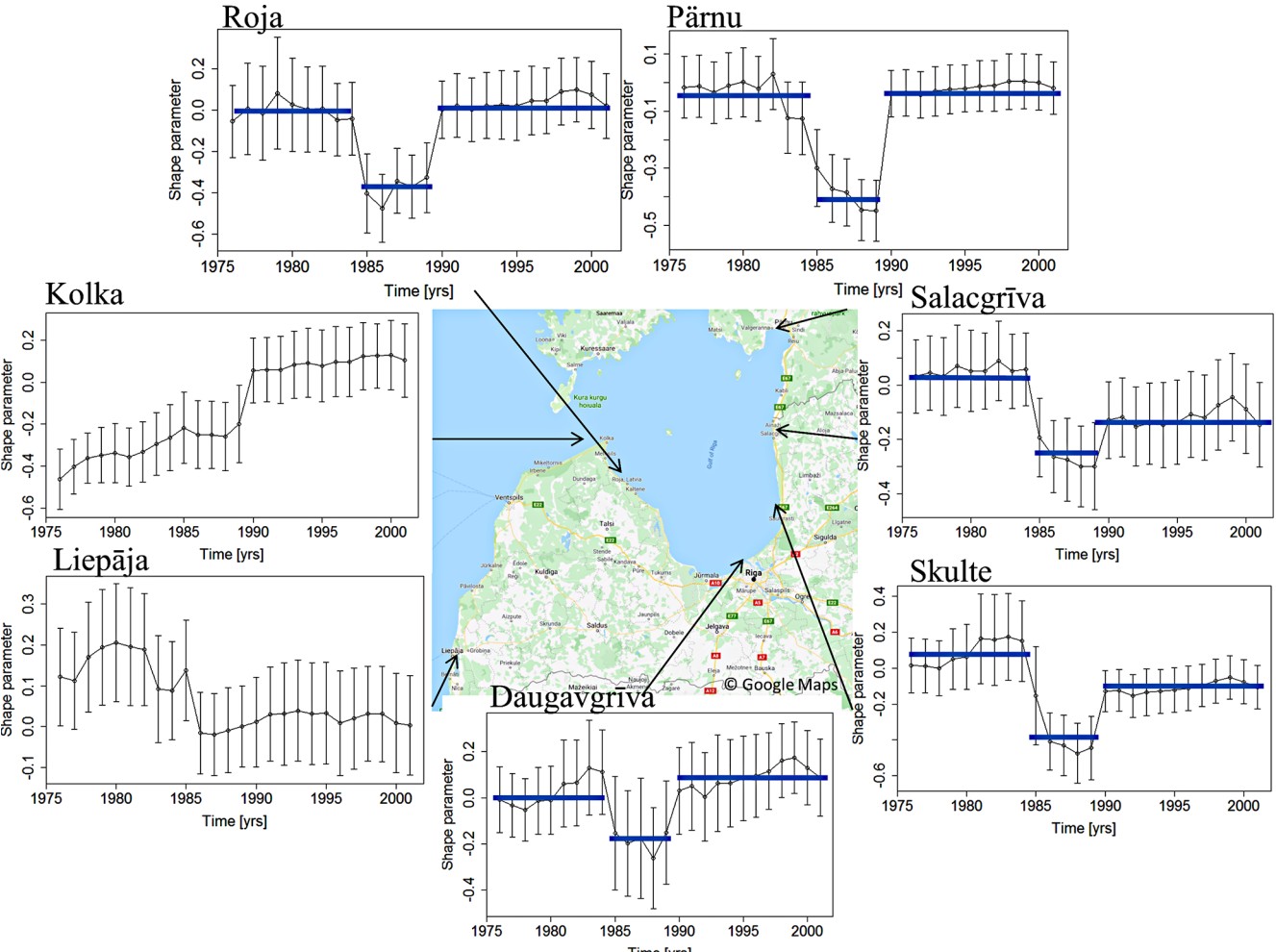

**Figure 7.** Shape parameter $\xi$ as a function of time for six tide gauge locations in Latvian waters and Pärnu in the northeastern Gulf of Riga. The presented estimates of the shape parameter were derived using a sliding-GEV fit with a window of 30 years. The time on the plots corresponds to the time in the middle of the sliding window. The blue lines show average values of the shape parameter for three time periods: before 1985, 1985–1989, and after 1989 (Table 2).

using 15 years before and after the listed "central" year of the fit, the regime shift may have occurred at a somewhat different instant in time.

To more consistently estimate the relative magnitude of the identified step-like changes with respect to the typical level of variations in the shape parameter, we make use of the overall appearance of the course of the shape parameter of the GEV distribution that resembles a step-like behavior with two instants of a regime shift. Using the estimated uncertainties $\sigma_e$, we can assess the statistical significance of these variations. The abrupt changes range in significance from $2\sigma_e$ (Daugavgrīva) to $7.8\sigma_e$ (Pärnu). The magnitude of the change at other stations is in the range of $4$–$5\sigma_e$. This shows a high significance of the shift at all stations except for Daugavgrīva. The described significant change in the shape parameter $\xi$ indicates a dramatic shift in the overall appearance of the extreme-value distribution from an approximately

Gumbel one to a Weibull-like shape in 1985 and then back to a Gumbel distribution in 1989.

Out of five locations which exhibit an abrupt shift in the shape parameter (Fig. 7), two tide gauges (Roja and Salacgrīva) exhibit a low level of completeness of about 30 % (Table 1). To test how this might affect the results of the extreme-value analysis, we introduced gaps to the sea level time series of one of the most complete stations, Pärnu. For that, 70 % of the data were randomly removed from the Pärnu data 100 times, and then precisely the same method of the non-stationary-extreme-value analysis was applied, and the presence of the step function in the shape parameter was assessed. The test showed that lower completeness leads to an underestimation of the location parameter. However, in 85% of the cases, the same abrupt shift in the shape parameter was observed. This indicates that in the case of the less complete

**Table 2.** The average value of shape parameter $\xi$ as a result of sliding-GEV fit with a window of 30 years before 1985, 1985–1989, and after 1989 (Fig. 7).

| Measurement site | $\xi$ (until 1985) | $\xi$ (1985–1989) | $\xi$ (from 1989) |
| --- | --- | --- | --- |
| Daugavgrīva | $0.03 \pm 0.06$ | $-0.2 \pm 0.1$ | $0.09 \pm 0.05$ |
| Pärnu | $-0.03 \pm 0.06$ | $-0.39 \pm 0.05$ | $-0.02 \pm 0.03$ |
| Roja | $-0.01 \pm 0.07$ | $-0.38 \pm 0.07$ | $0.04 \pm 0.05$ |
| Salacgrīva | $0.05 \pm 0.06$ | $-0.27 \pm 0.06$ | $-0.12 \pm 0.05$ |
| Skulte | $0.09 \pm 0.07$ | $-0.38 \pm 0.09$ | $-0.11 \pm 0.04$ |

stations, we do see the abrupt shift at an 85 % confidence level.

### 3.3  Non-stationary-extreme-value analysis

To model the non-stationary extreme values, we tested the following hypotheses:

a. The GEV location, scale, and shape parameters follow a linear trend (Eq. 4).

b. The GEV shape parameter follows a quadratic trend (Eq. 5).

c. The GEV shape parameter follows a step function (Eq. 6).

These model functions Eqs. (4–6) were used to describe, to a first approximation, the possible course of time variability in parameters in a non-stationary-GEV fit (Eq. 3). The linear trends (case a) are commonly used to describe the impact of climate change on the ocean or atmosphere conditions. The hypotheses (b) and (c) were chosen to reach a better description of the detected features of the variability in the shape parameter (Fig. 7).

The use of the assumption of the presence of a linear trend in all GEV parameters (case a) showed that only tide gauges located in the Baltic proper (Liepāja) or locations in the Gulf of Riga that are largely affected by the water level in the Baltic proper (Kolka) follow a tentative trend in at least one of the parameters (Fig. 8). The data from Kolka revealed a linear trend in the location parameter at a significance level of 89 % and in the shape parameter at a significance level of 94 %. The trend in the shape parameter follows the $\xi \approx -0.44 + 0.01\,(t - 1961)$ relation, where $t$ is time in years. The parameters of the GEV distribution for water level maxima at Liepāja tide gauge revealed weaker linear trends in the location parameter (81 % significance level) and scale parameter (84 % significance level). However, no linear trends with significance level > 80 % were detected in any of the GEV parameters at all sites in the interior of the Gulf of Riga. Even though the trends mentioned earlier at Liepāja and Kolka are not statistically significant at a commonly accepted 95 % level, the presented features indicate an intrinsic

difference in the behavior of water level extremes in the inner area of the Gulf of Riga compared to the stations that reflect water level in the Baltic proper. The 50-year return values calculated using the stationarity assumption (Fig. 8) also show a large difference between the inner area of the Gulf of Riga compared to the Baltic-proper area, indicating different dynamics of the water level extremes.

Interestingly, the results presented in Sect. 3.1 demonstrate that the sliding-GEV fit (with the assumption that the GEV parameters are constant within each single time window) showed almost no changes in the parameters of GEV at Liepāja. This confirms that sliding GEV can be used in assessing whether the extremes behave in a non-stationary manner. However, to accurately model the time variability in GEV, a more elaborate approach such as non-stationary-GEV fit, should be performed. A likely reason for this difference is that the sliding-GEV fit involves a smaller number of years within each window, whereas the non-stationary-GEV fit considers all available data.

The shape of temporal changes in various parameters (Figs. 5–7) suggests that even if a clear linear trend is missing, fittings using the more complicated functional shape of approximations may reveal further information about changes in these parameters. A quadratic function is a natural choice to highlight acceleration of changes as well as to identify the presence of cycles with periods longer than the entire time series if some of the parameters have a clearly defined minimum or maximum within the observation time interval. Fitting a quadratic trend to the shape parameter at all locations (case b) showed a significant fit only for the Skulte water level extremes, with a minimum in 1988 (at a significance level of 93 %). However, the fit did not follow well the observed change. Therefore, it is unlikely that the observed changes had a cyclic manner.

Finally, a step-like function was used to model the shape parameter changes following the sliding-GEV analysis that showed that the fitted shape parameters exhibit an abrupt shift (Fig. 7). To find a suitable approximation of such changes (case c, Eq. 6) which best describes the change and to specify when it happened and how long it lasted, we tested various settings of a step function (Eq. 6). The parameter $\Delta t$ was varied from 1 to 15 years. The corresponding total duration of the event ($2\Delta t$) was in the range of 2 to 30 years, correspondingly. The parameter $t_0$ corresponds to the time in the middle of the event. It was modified in the range of 1981–1991 with a step of 1 year. The amplitude of both regime shifts was assumed to be the same. It was changed from $-0.2$ to $-1$ incrementally with a step of 0.2. The constructed step functions were used to model the time variability in the GEV shape parameter at all locations. These functions were fitted to the set of block maxima of water levels using a non-stationary-GEV fit (Eq. 3).

The properties $\Delta t$ and $t_0$ of the best approximation using a step function for the course of the shape parameter vary broadly for individual stations. This does not necessar-

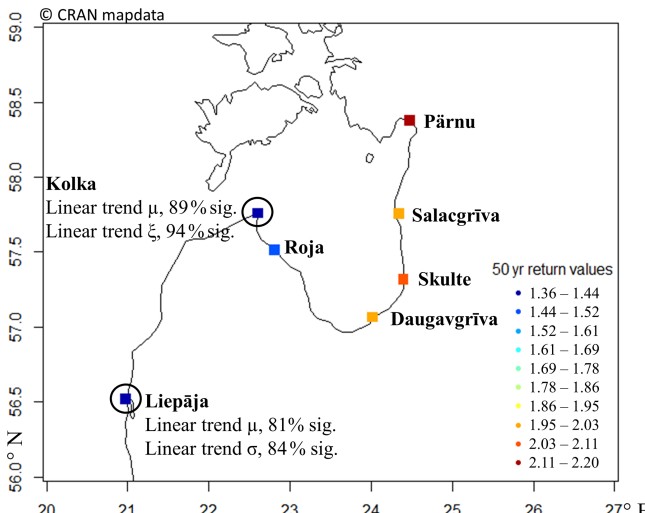

**Figure 8.** Fifty-year return values (color scale; numbers indicate the range in meters) obtained with a stationary-GEV function fit (squares). The tide gauges with a linear trend in one of the GEV parameters are marked with large black circles.

ily mean a large discrepancy of the properties of the relevant extreme-value distributions because recordings at different sites may be affected by several local features. To obtain the best fit for all stations, we used a collective location approach, where a sum of the goodness of fit for all stations was used as a measure of the quality of the fitted non-stationary distribution (e.g., Votier et al., 2008). This approach provided more consistent results. In some cases, however, it showed lower formal statistical significance compared to the tests performed for individual sites. The negative log-likelihood value of the non-stationary-GEV fit was used as a measure of the goodness of fit.

A sensitivity test was performed to check what parameters affected the fit the most. The goodness of fit was significantly affected by the change in $\Delta t$ and $t_0$. However, it was less sensitive to the amplitude of regime shift $a$ of the fitted step function in Eq. (6). Therefore, it was fixed to the value $a = 0.36$ that was averaged over all seven stations in Table 2. This feature indicates that the existing data are consistent in terms of the magnitude of the regime shift in the middle and end of the 1980s.

For the stations located in the interior of the Gulf of Riga (that is, excluding Kolka), the best fit for the shape parameter was $\Delta t = 2$, $t_0 = 1988$. The corresponding abrupt shift therefore formally started in 1986 and lasted until 1990. The likelihood ratio test showed that the non-stationary-GEV fit with the shape parameter following a step function with $\Delta t = 2$, $t_0 = 1988$ described extreme values better than the stationary fit. This claim is valid with statistical significance of 99.9 % at Roja and Salacgrīva, 98 % at Daugavgrīva, 85 % at Skulte, and 80 % at Pärnu. The identified step function is consistent with the sliding-GEV fit (Fig. 7, Table 2). However, the

formal statistical significance of the presence of an abrupt change depends on the method in use. In case of the sliding GEV, the most significant change was obtained at Pärnu, and the least significant one occurred at Daugavgrīva.

### 3.4 Links with large-scale climate indices

To study if the observed abrupt regime shift in the distribution of sea level extremes coincides with changes in the large-scale atmospheric circulation, we performed a correlation analysis between the sea level extremes and multiple teleconnections, such as Atlantic Multidecadal Oscillation (AMO), Atlantic Meridional Mode (AMM), Antarctic Oscillation (AAO), Arctic Oscillation (AO), East Atlantic (EA), East Atlantic–Western Russia (EATL–WRUS), North Atlantic Oscillation (NAO), Pacific–North American (PNA), Polar–Eurasia (Poleur), Scandinavia (SCAND), and Baltic Winter index (WIBIX), and compared the level of the correlation during the 1985–1990 period.

The WIBIX (Hagen and Feistel, 2005) was obtained from the Leibniz Institute for Baltic Sea Research, Warnemünde (https://www.io-warnemuende.de/wibix.html, last access: 1 September 2020). The other climatic indices were obtained from the National Oceanic and Atmospheric Administration (NOAA). The EA, EATL–WRUS, Poleur, PNA, and SCAND were downloaded from https://www.cpc.ncep.noaa.gov/data/teledoc/telecontents.shtml (last access: 1 September 2020) and the others from https://psl.noaa.gov/data/climateindices/list/ (last access: 1 September 2020).

Due to previously reported strong seasonality of correlations between the Baltic Sea water levels and NAO (e.g., Jaagus and Suursaar, 2013), the correlation analysis was performed for both annual and monthly maxima. The monthly mean was subtracted from the monthly maxima and yearly mean from the annual maxima to create stationary time series in a statistical sense. Pearson's correlation was employed for the correlation analysis below. We also used the non-parametric Kendall's and Spearman's correlation coefficients to test if it affects the results. Although Kendall's correlation resulted in approximately $\sim 30$ % lower values, the time variability in the correlation stayed exactly the same and therefore did not affect the results of this paper. For some indices, such as WIBIX, only the total correlation was calculated since only yearly values of the index were available.

In general, the total correlation coefficients, calculated between the annual maxima and annual climatic indices, exhibited low values. The AAO index had a negative correlation of $-0.4 \pm 0.2$ at Daugavgrīva (99 % significance level, denoted below as sign. l.) and $-0.3 \pm 0.2$ at Skulte (94 % sign. l.). The extreme water levels at Liepāja showed a low but significant positive correlation of $0.3 \pm 0.2$ with both NAO (99 % sign. l.) and AO (97 % sign. l.) indices. The other locations did not show any significant total correlations with any other studied indices. However, a sliding-correlation approach applied to the monthly sea level maxima and the climatic indices re-

vealed significant changes in the correlation coefficients in time and with seasons. Figure 9 shows running-correlation results with a window of 15 years (only the correlations with > 95th percentile confidence levels are shown) of Daugavgrīva water level extremes with NAO, SCAND, and PNA teleconnections. The indices in Fig. 9 revealed a significant change in the correlation during the 1986–1990 period of the abrupt regime shift. The Arctic Oscillation showed practically the same results as the NAO, which was typical for the Baltic Sea region (Jaagus and Suursaar, 2013).

The sliding-correlation analysis revealed that the NAO, AO, and SCAND indices showed remarkably similar results for all the stations in the interior of the Gulf of Riga (Daugavgrīva, Roja, Pärnu, Salacgrīva, and Skulte). For all these locations, the NAO index was characterized by weakening of the correlation during the 1985–1990 period and shift in seasonality (Fig. 10). Before 1984, the NAO index showed the highest correlation in January. However, after 1988, the highest correlation coefficients were observed in March (Fig. 10), exhibiting an abrupt change in seasonal correlation with the NAO. Therefore, the non-stationarity of sea level extremes in the area is most likely caused by the severe time variability in the NAO signal in the sea level extremes.

The SCAND index, on the other hand, revealed the strongest negative correlation in March during 1982–1990 consistently for the whole interior of the Gulf of Riga stations. However, the Liepāja and Kolka tide gauges exhibited different dynamics of the sliding correlation with these teleconnections. The correlation with NAO revealed a gradual positive trend in Liepāja and a negative trend in Kolka (Fig. 10). This change in correlation with NAO can explain the observed reversed changes in the GEV shape parameters in Liepāja and Kolka (Fig. 7) since a stronger correlation with NAO corresponds to more frequent and powerful storms.

Interestingly, the time series from Daugavgrīva and Skulte tide gauges showed a significant correlation (> 95 % significance) with the PNA teleconnection in December and October (Figs. 9 and 11). The correlation coefficients reached $0.6 \pm 0.3$ (December) and $0.5 \pm 0.3$ (October) at Daugavgrīva during 1984–1988. Since the El Niño and PNA can in principle affect the European climate intermittently and in a non-stationary way (e.g., Brönnimann, 2007), it is possible that during the weakening of the correlation with the NAO, a faint signal from the other factors affecting the European climate was detected in this study. However, a more detailed analysis is required to study the Pacific region's effects on the Baltic Sea climate.

## 4   Conclusions and discussion

The core conclusion from the performed analysis is that the parameters of theoretical extreme-value distributions of the observed water level maxima in Latvian waters, both on the shore of the Baltic proper and in the interior of the Gulf of

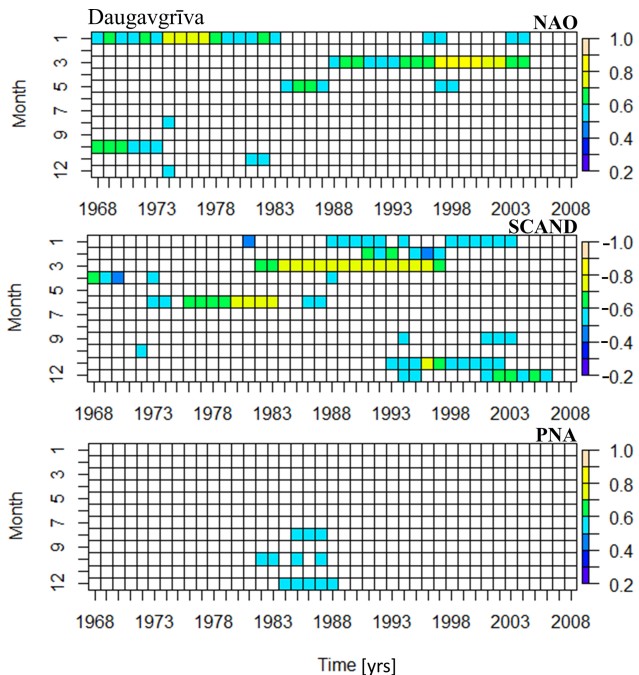

**Figure 9.** Sliding-correlation results for Daugavgrīva and NAO, SCAND, and PNA climatic indices. The correlation is calculated for the water level monthly maxima with a window of 15 years. Only the correlation coefficients with a significance of more than 95 % are shown.

Riga, exhibit statistically significant changes over the last 60 years. The most remarkable changes occur in the shape parameter of the GEV distribution. These changes may cause fundamentally different projections for long-term behavior of water level extremes and their return periods as the underlying extreme-value distribution changes from a three-parameter (reversed) Weibull distribution to a Gumbel one. While the reversed Weibull distribution has a finite upper limit, any water level height may occur according to Gumbel distributions.

Surprisingly, the nature of changes in the shape parameter of the GEV distribution for water level maxima is substantially different on the shores of the Baltic proper and in the interior of the Gulf of Riga. This parameter changes more gradually and mostly in one direction at sites directly affected by the water level regime of the Baltic proper. The direction of changes is different at different locations: the shape parameter decreases on the open shore of the Baltic proper at Liepāja but increases at Kolka.

Interestingly, the temporal course of the shape parameter has a pair of regime shifts at all measurement sites in the interior of the Gulf of Riga. During most of the time, this parameter is close to 0, and, therefore, the GEV distribution can be approximated well by a Gumbel distribution. The shape parameter becomes negative (and thus a reversed three-parameter Weibull distribution governs the water level

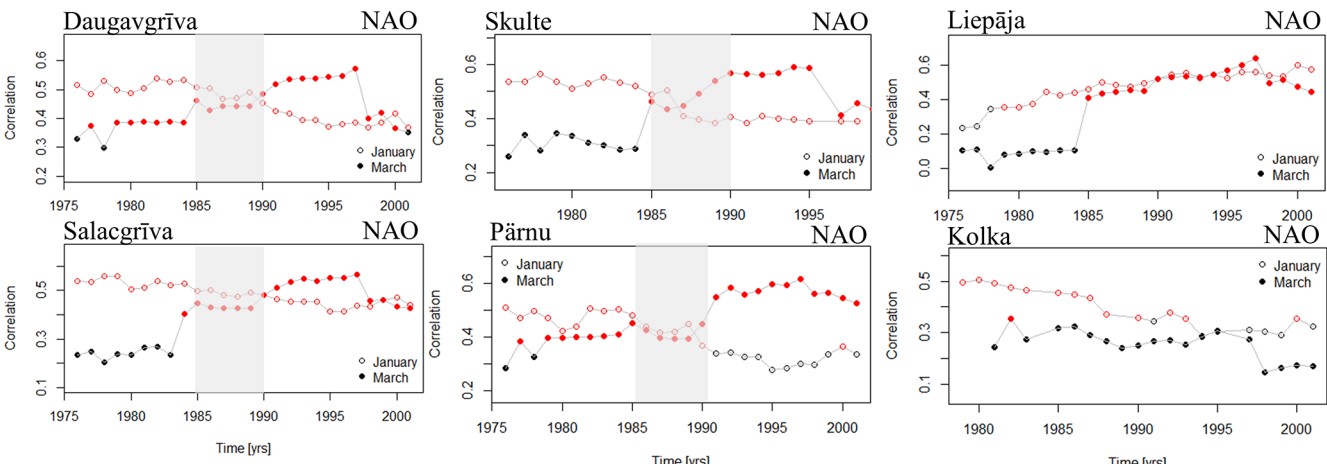

**Figure 10.** Sliding-correlation results between the water level maxima in January (open circles) and March (solid circles) with the NAO index. The correlation is calculated using a window of 30 years. The correlation coefficients with a significance of more than 95 % are marked in red. The gray rectangle highlights the period of the abrupt regime shift.

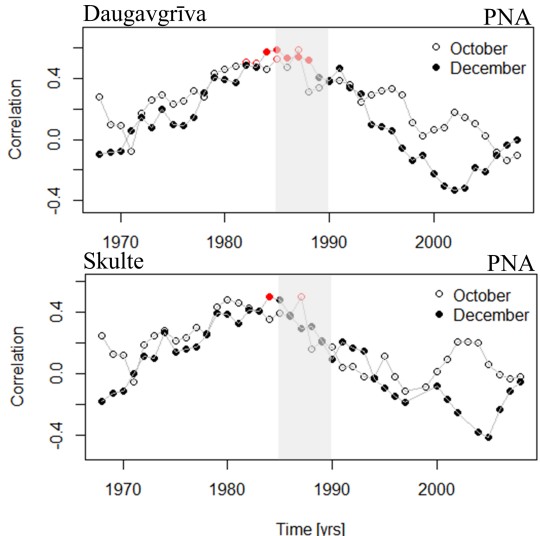

**Figure 11.** Sliding-correlation results between the water level maxima in October (open circles) and December (solid circles) with the PNA index. The correlation is calculated using a window of 15 years. The correlation coefficients with a significance of more than 95 % are marked in red. The gray rectangle highlights the period of the abrupt regime shift.

maxima) around the year 1986. In contrast, its value jumps back to a level close to 0 around the year 1990. This regime shift may reflect abrupt changes in geostrophic winds in 1987/1988 (Soomere et al., 2015b) and surface-level winds in the 1980s (Keevallik and Soomere, 2014). The described process may also mirror massive evidence of regime shift in various abiotic variables in Estonia in 1989–1990 in biotic time series of bogs and marine ecosystems in 1990 (Kotta et al., 2018).

We demonstrate that the period of an abrupt shift (1986–1990) in the shape parameters of the GEV distribution in the interior of the Gulf of Riga coincides with the weakening of the correlation with NAO. However, the tide gauges at the Latvian shore of the Baltic proper showed completely different time variability in the link with the NAO, showing an increase in case of Liepāja and a decrease in Kolka (Fig. 10). The different appearance of changes to the shape parameter and nature of correlation with NAO in the interior of the Gulf of Riga and at the Latvian shore of the Baltic proper suggests that there is a strong relationship between the shape parameter of the GEV distribution and correlation with NAO in the Baltic Sea. The difference in the temporal course of the shape parameter in the two basins may stem from the very nature of the formation of extreme water levels in the Gulf of Riga. It is a multi-step process that is possibly only effective under a specific sequence of atmospheric events. The probability of occurrence of such events is evidently correlated with the values of the NAO index.

Many previous works have found significant correlations between winter sea level changes (that usually are responsible for the extreme water levels) and changes in wind and air pressure as well as with the North Atlantic Oscillation (NAO) index in the Baltic Sea in general (e.g., Andersson, 2002) and in Pärnu, in the Gulf of Riga, in particular (Suursaar and Sooäär, 2007). Similar links have been established for the neighboring areas. For example, the NAO contributes to changes in the mean and extreme water levels in the North Sea (Tsimplis et al., 2005). The link between the NAO and the Baltic Sea level has displayed substantial decadal variations in the last 2 centuries; however, the link was perceived as spatially heterogeneous, even in wintertime (e.g., Andersson, 2002; Jevrejeva et al., 2005; Hünicke et al., 2015; Karabil et al., 2017). In this paper, we demonstrated that there is

a significant spatial difference in the link between the Baltic Sea extreme water levels and NAO.

The established connection between the time variable link with the NAO and the non-stationary behavior of the GEV parameters of extreme water levels implies that if the other regions in the world exhibit a time-variable link with atmospheric teleconnections, it can result in non-stationary behavior of the sea level extremes and under- or overestimation of the risks.

*Code and data availability.* Code is available from the authors on request (MATLAB and R scripts). The tide gauge data are provided by the Latvian Environment, Geology and Meteorology Centre (LEGMC) and are available at http://www.meteo.lv (last access: 8 March 2021) (Ūdeņu monitoringa programma, 2020).

*Data availability.* The tide gauge data are provided by the Latvian Environment, Geology and Meteorology Centre (LEGMC) and are available for download at https://www.meteo.lv/hidrologija-datu-pieejamiba/ (last access: 1 July 2020) (Ūdeņu monitoringa programma, 2021).

*Author contributions.* NK designed the study; developed the R scripts; performed non-stationary-GEV fits and correlation analysis with the climatic indices; produced most of the figures; wrote Sect. 2.4, 3, and parts of the introduction and discussion; and checked the consistency of the results. TS compiled most of the parts of the introduction and discussion and checked the consistency of the results. RM checked the tide gauge data quality; developed the MATLAB scripts; wrote Sect. 2.1, 2.2, and 2.3; and produced Figs. 1, 2, and 4.

*Competing interests.* The authors declare that they have no conflict of interest.

*Special issue statement.* This article is part of the special issue "Advances in extreme value analysis and application to natural hazards". It is a result of the Advances in Extreme Value Analysis and application to Natural Hazard (EVAN), Paris, France, 17–19 September 2019.

*Acknowledgements.* The authors are deeply grateful to the Latvian Environment, Geology and Meteorology Centre and the Estonian Weather Service for making the water level data publicly available. We thank the anonymous referees for the thoughtful comments.

*Financial support.* This research has been supported by the Estonian Research Council (grant nos. IUT33-3 and PRG1129 and Mobilitas project MOBTT72) and the European Economic Area

(EEA) Financial Mechanism 2014–2021 Baltic Research Programme (grant no. EMP480).

*Review statement.* This paper was edited by Thomas Wahl and reviewed by two anonymous referees.

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
