# Peer review of "Non-stationary analysis of water level extremes in Latvian waters, Baltic Sea, during 1961–2018."

_Natural Hazards and Earth System Sciences, 2020_

## Referee Comment (RC1) · Anonymous Referee #1 · 14 May 2020

NHESS-2020-100 review report:

Non-stationary analysis of water level extremes in Latvian waters, Baltic Sea, during 1961–2018. Authors: N. Kudryavtseva, T. Soomere, R. Männikus

This paper deals with analysis of water level extremes in the eastern Baltic Sea using various statistical procedures (GEV, non-stationary and regime shift analysis). The topic is significant in the era of climate change and sea level rise. The paper seems to be an extension to some recent papers by the same authors (e.g.: Identification of mechanisms that drive water level extremes from in situ measurements in the Gulf of Riga during 1961–2017 by Männikus, Soomere, Kudryavtseva in Cont Shelf Res 2019;

[Figure]

Non-stationary Modeling of Trends in Extreme Water Level Changes Along the Baltic Sea Coast by Kudryavtseva, Pindsoo, Soomere in J Coastal Res Special Issue 2018; Variations in parameters of extreme value distributions of water level along the eastern Baltic Sea coast by Soomere, Eelsalu, Pindsoo in Estuarine Coast Shelf Res, 2018). Nevertheless, the new article includes plenty of new insight into the subject of the Gulf of Riga water level maxima and therefore deserves publication.

The paper is well written, adequately organized and well-illustrated with figures. I suggest publication with medium to minor corrections.

Comments

1) Please clarify the study area. In L. 83-85 it is said that the study focusses on Gulf of Riga. A bit later (line 97) it said that "The study area – the shores of Latvia with a total length of about 500 km..." I think that neither is fully correct: Besides Gulf of Riga, also Latvian coasts in the Baltic Proper are considered. Secondly, large part of the discussed Gulf of Riga actually belongs to Estonia, including one station where the data come from (Pärnu). Also the paper title says that the study area is "Latvian waters". I understand that it is difficult to conveniently introduce such details in the title. But the fact about Pärnu and Estonia should be stated possibly earlier in the paper. Currently it is hidden to far quarters of the manuscript.

2) L.110 "...may increase the average sea level in the entire Baltic Sea by almost 1 m for several months (Soomere and Pindsoo, 2016)". I believe this 1 m is exaggerated for the entire Baltic. Soomere and Pindsoo (2016) said "...raise the average sea level by almost 1 m for a few weeks". Weeks - and it was probably for the eastern part of the sea. But then, sea level must be lower in other parts. Johansson and Kahma (Boreal Env Res, 2016, 21), say on p.34 with monthly-based analysis: "The Baltic Sea average level Hd ranged from –43 to +51 cm in 1933–2012". One month 51 cm, one month 45-50 cm. L.118 Considering the above-said, also this 1+1+1 m quantification does not quite hold.

[Figure]

3) L.130, Table 1. I wonder, in three stations the hourly data completeness is 30% and in other three 99%. Please ensure shortly that it does not influence the statistics (distribution parameters) – particularly considering the potential completeness changes over years in these 30% stations.

4) Were the data 1961-2018 detrended before analysis, or perhaps, it is not a problem? I.e., is there a possible influence of local uplift/subsidence or global sea level rise to e.g. variations in location parameter?

5) L. 320- Major regime shift in the Baltic region in 1989/90 is well known indeed. However, how to explain this odd shift in the form of "dent" in the shape parameter inside the Gulf of Riga in 1984-90? Seems like an artefact. In L.430- it is shortly discussed on the basis of changes in average air flow speed (Keevallik and Soomere 2014), possibly including regime shifts in 1987 (up) and 1993 (down) above the Gulf of Finland. How about changes in wind direction? For instance, Fig.7e,g,h by Suursaar (http://dx.doi.org/10.3176/earth.2013.05) shows variations in annual resulting wind direction at stations near Gulf of Riga. In 1985-94, the direction was quite stable 230-250 deg (good direction for inflow through Irbe Strait), while before and after it fluctuated plus-minus off that direction. Just a guess.

6) Fig.8. If the numbers in brackets mark the range in centimetres (134,141), why not express it 136-141? What does this range mean anyway - neither explained in the text nor in legend. What does Pärnu 50-yr return value 211,220 mean? Pärnu had measurements 253 cm and 275 cm within 39 years.

Minor/technical comments

L.23 too many (9) keywords, some repeating title words

L.31 This highligh "Best fit in terms of step-like approximation for shape parameter established" makes no sense as a highlight. Also the third highlight misses something.

L.39 Mudersbach and Jensen

[Figure]

L.58 easternmost Baltic Sea – feels weird

L.78 open coast of Latvia – feels weird

L.105 (100, etc) Väinameri (Moonsund) sub-basin in the Western Estonian archipelago –> West Estonian Archipelago Sea

Please unify units in the Fig. axes (commas, parentheses): Water level, cm; Return period, year; Time [yr]; Time [yrs] etc.

L.205 hrs – not much shorter than hours

L.443 Klevanny -> Klevannyy

---

## Referee Comment (RC2) · Anonymous Referee #2 · 4 Jun 2020

I have read the MS with great interest, the paper is well-written, and the organization is good. The study provides updated knowledge of extreme water levels in the Gulf of Riga, which seems to be significantly different from those found in the Baltic Sea. It is also interesting the temporal behavior of the shape parameter in those tide gauges located in the interior of the Gulf.

Despite I operate outside this study area, it seems like many efforts have already been done in the literature to explain changes in mean sea level, extreme sea levels, and distribution parameters in the Baltic Sea and the Gulf of Riga. I, therefore, believe that the paper would require more work to create additional value to the existing literature,

for example, by extending the study to the possible causes of the temporal changes in the tail distribution of the extreme water levels.

Please also note the supplement to this comment: https://www.nat-hazards-earth-syst-sci-discuss.net/nhess-2020-100/nhess-2020-100-RC2-supplement.pdf

―――――――――――――――――

[Figure]

**Supplement:**

**Review of "Non-stationary analysis of water level extremes in Latvian waters, Baltic Sea, during 1961–2018."**
*Nadezhda Kudryavtseva, Tarmo Soomere, Rain Männikus*

This paper explores the temporal changes in the parameters of the General Extreme Value (GEV) distribution for water level extremes in the Gulf of Riga and Liepaja (eastern Baltic Sea). Temporal changes in GEV parameters are assessed for the seven locations by fitting a stationary GEV distribution to observed sea level data separated into a 30-year sliding window. Also, a non-stationarity GEV distribution is fitted to the water level extreme data.

Results show that changes in the GEV parameters, as well as the significance level of these changes, vary across sites, with higher differences when comparing those stations located in and out the Gulf of Riga. Consistent results are obtained for changes in the shape parameter at those sites located in the interior of the Gulf of Riga: a significant drop around 1986 and a posterior return by 1990. A step-function is fitted to these water level extremes, obtaining similar results. This behavior is not found at those sites affected by the water level in the Baltic Sea (Kolka and Liepāja). For these, some GEV parameters are found to follow a linear trend: location and shape in Kolka and location and scale in Liepāja.

**General comment**

I have read the MS with great interest, the paper is well-written, and the organization is good. The study provides updated knowledge of extreme water levels in the Gulf of Riga, which seems to be significantly different from those found in the Baltic Sea. It is also interesting the temporal behavior of the shape parameter in those tide gauges located in the interior of the Gulf.

Despite I operate outside this study area, it seems like many efforts have already been done in the literature to explain changes in mean sea level, extreme sea levels, and distribution parameters in the Baltic Sea and the Gulf of Riga. For instance, changes in extreme events as well as changes in extreme value distributions have been studied at Parnü (Suursay 2007, Eelsalu 2014, Ekman 1996, among others). Also, spatial variations in extreme sea level distributions have been studied along the eastern coast of the Baltic Sea (Soomere 2018, Ekman 1996). I, therefore, believe that the paper would require more work to create additional value to the existing literature, for example, by extending the study to the possible causes of the temporal changes in the tail distribution of the extreme

water levels. For instance, previous works have found significant correlations between winter months sea level changes and changes in wind and pressure changes as well as with the North Atlantic Oscillation index in the Baltic Sea and at Parnü (Anderson 2002, Suursay 2007, among others). Although in a different study area, Tsimplis et al. (2005) found that NAO contributes both to changes in the mean and extreme water levels in the North Sea. Correlation between climate indexes and GEV distribution parameters were previously assessed in other areas (for instance, Méndez et al 2006).

**Other comments**

- Sometimes in the text, Fig and Figure are used indistinctly.

- Line 190. Figure 4 is called before figure 3 in the main text.

- Line 220/225. In order to assure reproducibility, I would recommend indicating the equations, functions, or tools used in Matlab as well as a quick description of *Hydrognomon*, for those who are not familiar with it.

- In the same sense, I recommend an extension of the explanation about the water level data processing before performing the distribution fitting. By doing so, the reader can reproduce and understand how the data have been processed to obtain the extreme water level from the tide gauge records.

- Figures. The overall quality of the figures should be improved, for instance, by matching the font size and font name with the main text.

- Line 260. I would recall here, as the authors do in section 2.3, that changes in the shape parameter have consequences in the tail of the distribution, determining the behavior of the events with very low frequency.

- Line 270, figure 5. It would be good to see results for all other locations, as in figure 7. Same for the results of scale parameters (figure 6).

- Line 290. *"As the 95% confidence intervals of estimates of this parameter for single years mostly overlap, it is safe to say that the location parameter of the GEV distribution of water level extremes has not experienced any substantial changes in Latvian waters since the 1960s".* I'm curious about changes in the location parameter at Skulte, these changes appear significant when comparing the location parameter obtained by ~2000 vs ~1975. Also, including a grid in the figures would improve the comparison across years. I recommend also to further explain how the statistical significance has been calculated.

- Line 415. *"… observed and measured water maxima in…"*. Aren't they the same?

Andersson H.C. (2002) Influence of long-term regional and large-scale atmospheric circulation on the Baltic sea level, Tellus A: Dynamic Meteorology and Oceanography, 54:1, 76-88, DOI: 10.3402/tellusa.v54i1.12125

Eelsalu M., Soomere T., Pindsoo K., Lagemaa P. (2014) Ensemble approach for projections of return periods of extreme water levels in Estonian waters: Continental Shelf Research, 91, 201-210, DOI: 10.1016/j.csr.2014.09.012

Ekman M. (1996) Extreme annual means in the Baltic Sea level during 200 years: Small Publ. Hist. Geophys. 2, 15 pp.

Méndez F. J., Menéndez M., Luceño A. (2007) Analyzing Monthly Extreme Sea Levels with a Time-Dependent GEV Model: Journal of atmospheric and oceanic technology, 24, DOI: 10.1175/JTECH2009.1.

Soomere T., Eelsalu M., Pindsoo K. (2018) Variations in parameters of extreme value distributions of water level along the eastern Baltic Sea coast: Estuarine, Coastal and Shelf Science (2018), DOI: 10.1016/j.ecss.2018.10.010.

Suursaar Ü. & Sooäär J. (2007) Decadal variations in mean and extreme sea level values along the Estonian coast of the Baltic Sea, Tellus A: Dynamic Meteorology and Oceanography, 59:2, 249-260, DOI: 10.1111/j.1600-0870.2006.00220.x

Tsimplis M.N., Woolf D. K., Osborn T. J., Wakelin S., Wolf J., Flather R., Shaw A. G. P., Woodworth P., Challenor P., Blackman D., Pert F., Yan Z., Jevrejeva S. (2005) Towards a vulnerability assessment of the UK and northern European coasts: the role of regional climate variability: Phil. Trans. R. Soc. A, 363, 1329–1358, DOI:10.1098/rsta.2005.1571

---

## Author Comment (AC1) · 31 Jul 2020

We thank the referee for careful reading the manuscript, critical assessment of the analysis and useful and professional comments which helped to improve the paper. Please find below the detailed replies to all the questions.

1) Please clarify the study area. In L. 83-85 it is said that the study focusses on Gulf of Riga. A bit later (line 97) it said that "The study area – the shores of Latvia with a total length of about 500 km..." I think that neither is fully correct: Besides Gulf of Riga, also Latvian coasts in the Baltic Proper are considered. Secondly, large part of the discussed Gulf of Riga actually belongs to Estonia, including one station where

[Figure]

the data come from (Pärnu). Also the paper title says that the study area is "Latvian waters". I understand that it is difficult to conveniently introduce such details in the title. But the fact about Pärnu and Estonia should be stated possibly earlier in the paper. Currently it is hidden to far quarters of the manuscript.

- Thank you for this observation. The study area involves all Latvian waters, but we also use data from one Estonian tide gauge. We state now this fact earlier in the paper.

2) L.110 "...may increase the average sea level in the entire Baltic Sea by almost 1 m for several months (Soomere and Pindsoo, 2016)". I believe this 1 m is exaggerated for the entire Baltic. Soomere and Pindsoo (2016) said "...raise the average sea level by almost 1 m for a few weeks". Weeks - and it was probably for the eastern part of the sea. But then, sea level must be lower in other parts. Johansson and Kahma (Boreal Env Res, 2016, 21), say on p.34 with monthly-based analysis: "The Baltic Sea average level Hd ranged from –43 to +51 cm in 1933–2012". One month 51 cm, one month 45-50 cm. L.118 Considering the above-said, also this 1+1+1 m quantification does not quite hold.

- Thank you, this was an inexact formulation. We had in mind that large water volumes may increase the water level by 1 m within a few weeks. This increase usually starts from a lower than average water level. Still, Pindsoo and Soomere 2016 have shown that water levels elevated by 60–80 cm over long-term average have persisted for several weeks in the eastern Baltic Sea. We have formulated this aspect carefully now in the manuscript. The increase in water level in the eastern Gulf of Riga by more than 1 m compared to the water level at the Baltic proper shores of Latvia has occurred several times since the 1960s (Männikus et al., 2019). It is natural that the 1+1+1 m process does not work as a simple sum as conditions favourable for one mechanism are not perfect for others.

3) L.130, Table 1. I wonder, in three stations the hourly data completeness is 30% and in other three 99%. Please ensure shortly that it does not influence the statistics

(distribution parameters) – particularly considering the potential completeness changes over years in these 30% stations.

- Although some of the tide gauges had lower completeness, it was uniformly low except for the period after 2005 there was an increase in a number of measurements. None of the stations showed a change in completeness during the years of the abrupt shift. We performed a test to check how low completeness could have affected the results. For that, we took one of the most complete stations, Pärnu, randomly removed 70% of the data and applied precisely the same method of the extreme value analysis. The resulted data had 30% completeness, which is characteristic for the less complete stations, such as Roja and Salacgriva. Running 100 times the same analysis, we found that the same abrupt change in the shape parameter was observed even with only 30% completeness in 85% of the cases. This indicates that in case of the less complete stations, we do see the abrupt shift at an 85% confidence level. However, the tests showed that lower completeness leads to the location parameter being significantly underestimated. An explanation about how the completeness of the data could affect the extreme value distribution fitting is added to the paper.

4) Were the data 1961-2018 detrended before analysis, or perhaps, it is not a problem? I.e., is there a possible influence of local uplift/subsidence or global sea level rise to e.g. variations in location parameter?

- The signal of the mean sea level variations was removed from the extremes. This removes all slowly varying sea level changes, and the analysis focuses only on the storm surges. Therefore, the influence of the local uplift or global sea level rise can not affect the results of the manuscript.

5) L.320- Major regime shift in the Baltic region in 1989/90 is well known indeed. However, how to explain this odd shift in the form of "dent" in the shape parameter inside the Gulf of Riga in 1984-90? Seems like an artefact. In L.430- it is shortly discussed on the basis of changes in average air flow speed (Keevallik and Soomere 2014),

possibly including regime shifts in 1987 (up) and 1993 (down) above the Gulf of Finland. How about changes in wind direction? For instance, Fig.7e,g,h by Suursaar (http://dx.doi.org/10.3176/earth.2013.05) shows variations in annual resulting wind direction at stations near Gulf of Riga. In 1985-94, the direction was quite stable 230-250 deg (good direction for inflow through Irbe Strait), while before and after it fluctuated plus-minus off that direction. Just a guess.

- The problem is that interrelations between wind properties at a single location and extreme water levels are not straightforward. Extreme water levels in this part of the world are created by long sequences of "events" that affect water surface at different locations. The background for the largest extremes (elevated water level of the entire sea) is created by specific atmospheric forcing in the southwestern Baltic Sea. Pumping of water into the Gulf of Riga requires another subsequent forcing pattern and high local surge, possibly the third pattern. The analysis of (Männikus et al., 2019) reveals a recognizable change in the directional structure of strong winds at Vilsandi. We hope to look in more detail into the local wind patterns in the future, but it is currently out of the scope of this manuscript. However, we added a section to the revised version of the manuscript describing the connection between the extreme water levels in the Latvian waters with multiple climatic indices (please check the reply to the referee 2). This shows that the observed phenomenon has a weak relation to the local wind and stronger connection to the global atmospheric circulation.

6) Fig.8. If the numbers in brackets mark the range in centimetres (134,141), why not express it 136-141? What does this range mean anyway - neither explained in the text nor in legend. What does Pärnu 50-yr return value 211,220 mean? Pärnu had measurements 253 cm and 275 cm within 39 years.

- Thank you, the numbers in brackets were changed to the range as advised. The 50-yr return value of Parnu in the range 211-220 cm was obtained by removing the slowly varying components from the data, then the generalized extreme value distribution (stationary in case of Fig. 8) was fitted to all the yearly maxima, and the return value

was calculated. These measurements indeed showed a maximum of 275 cm in Parnu on 9th of January 2005. After removing the mean variation from this extreme, it is only 269 cm. However, this extreme event was only one of a kind. Fitting a distribution considers all the extremes and underestimated this outlier. The outliers are discussed in detail in section 2.3. The corresponding explanation was added to the text.

Minor/technical comments

L.23 too many (9) keywords, some repeating title words

- We reduced the number of keywords.

L.31 This highligh "Best fit in terms of step-like approximation for shape parameter established" makes no sense as a highlight. Also the third highlight misses something.

- We modified the formulation of this highlight

L.39 Mudersbach and Jensen

- We corrected the typo

L.58 easternmost Baltic Sea – feels weird

- We changed to "eastern subbbasins of the Baltic Sea."

L.78 open coast of Latvia – feels weird

- We changed to "Baltic proper shores of Latvia."

L.105 (100, etc) Väinameri (Moonsund) sub-basin in the Western Estonian archipelago –> West Estonian Archipelago Sea

- We added this change

Please unify units in the Fig. axes (commas, parentheses): Water level, cm; Return period, year; Time [yr]; Time [yrs] etc.

- The units are unified

L.205 hrs – not much shorter than hours

- The notation for hours is written in a unified way now

L.443 Klevanny -> Klevannyy - Thank you; this name appears differently in different sources.

Please also note the supplement to this comment:
https://nhess.copernicus.org/preprints/nhess-2020-100/nhess-2020-100-AC1-supplement.pdf

---

## Author Comment (AC2) · 31 Jul 2020

General comment

Despite I operate outside this study area, it seems like many efforts have already been done in the literature to explain changes in mean sea level, extreme sea levels, and distribution parameters in the Baltic Sea and the Gulf of Riga. For instance, changes in extreme events as well as changes in extreme value distributions have been studied at Pärnu (Suursaar 2007, Eelsalu 2014, Ekman 1996, among others). Also, spatial variations in extreme sea level distributions have been studied along the eastern coast of the Baltic Sea (Soomere 2018, Ekman 1996). I, therefore, believe that the paper

would require more work to create additional value to the existing literature, for example, by extending the study to the possible causes of the temporal changes in the tail distribution of the extreme water levels. For instance, previous works have found significant correlations between winter months sea level changes and changes in wind and pressure changes as well as with the North Atlantic Oscillation index in the Baltic Sea and at Pärnu (Anderson 2002, Suursaar 2007, among others). Although in a different study area, Tsimplis et al. (2005) found that NAO contributes both to changes in the mean and extreme water levels in the North Sea. Correlation between climate indexes and GEV distribution parameters were previously assessed in other areas (for instance, Méndez et al 2006).

Reply: We thank the referee for careful reading the manuscript, critical assessment of the analysis and useful and professional comments which helped to improve the paper. We fully agree with this observation and recommendations. There are many studies into water level extremes in the Baltic Sea in general. Also, many issues of the water level regime have been addressed in the context of Estonian, Russian (Gulf of Finland) and Finnish waters. Strangely, such studies are missing for the Latvian waters (except for a recent one by ourselves [Männikus et al., 2019]). For this reason, it seemed to us that the almost unexplored Latvian water level data deserve focused attention. Most of the existing studies into extreme water levels in the Baltic Sea basin have been performed under the assumption of stationarity of the underlying extreme value distributions. Only Suursaar and Sooäär (2007) ask the question of possible changes in the parameters of the GEV distribution. However, they choose basically the same data set (1923-2005) and exclude the maximum in 2005 and then also the maximum in 1967. Eelsalu et al. (2014) address the possibilities of ensemble approach for the construction of a better projection of extreme water levels and their return periods, and for the identification of locations with a substantial magnitude of local effects. Ekman (1996) provides an analysis of the extremes of the water volume of the Baltic Sea. Soomere et al. (2018) study spatial variations in the parameters of extreme value distributions. All this research is, to some extent, relevant in the context of our study

and has been mentioned in the original submission. We have extended the Introduction towards more precise coverage of the listed items and added (Ekman, 1996) into the list of references.

We also agree that much more research is necessary to attribute the changes and patters we highlight with the possible forcing factors. Thank you for reminding this issue and providing the relevant references. The link between the NAO and the Baltic Sea level is spatially very heterogeneous (even in wintertime) and has also displayed substantial decadal variations in the last two centuries (e.g. Andersson, 2002; Jevrejeva et al., 2005; Hünicke et al., 2015, Karabil et al., 2017). We would like to mention also that although the relation between the NAO and the mean water level was addressed in a multitude of papers, the relation to the sea level extremes is not studied in detail. Following the suggestion, we added a section to the paper describing the connection between the extreme water levels in the Latvian waters with the main driving climatic indices in the region. Overall, the relation between the sea level extremes in the Gulf of Riga and the climatic indices is highly unstable and significantly variable in time. The figure below shows the running correlation results (only the correlations with more than 95% confidence) of Daugavgriva water level extremes with NAO, SCAND, and PNA, which showed the strongest relation. The Arctic Oscillation showed the same results as the NAO and is not shown here. The other locations in the Gulf of Riga showed the same results. Using the running correlation analysis, we found that the relation with NAO was the weakest during the 1984-1988 period.

Moreover, during that period, a shift in the months most affected by NAO is observed. Before 1984, the NAO index showed the highest correlation in January. However, after 1988, there is an abrupt shift, and after that, the largest correlation coefficients were observed in March. Multiple teleconnections were studied using the correlation analysis, such as SCAND, AO, AAO, EA, EATL-WRUS, Poleur, EP-NP, PNA, PT, THN, WP and tested which one had shown the highest correlation during the 1985-1990 period. The highest (negative) correlation during that period is observed with the SCAND index. In

the Baltic Sea region, the positive phase of NAO is associated with the westerly winds and negative phase with more frequent winds from the east and north-east (e.g., Trigo, Osborn and Corte-Real 2002). The SCAND mode, on the other hand, is responsible for the south-easterly winds in the positive phase and north-westerly winds in the negative phase (Bueh and Nakamura 2007; Gao, Yu and Paek 2017). This indicates that the change in the running correlation coefficients from positive correlation with NAO to negative correlation with SCAND is most likely caused by the short-term change in the prevailing wind direction.

Interestingly, the second-highest correlation (corr. coefficient $\sim$0.55) during 1985-1990 is found between the water level extremes and PNA index (see the bottom panel of the figure below). The PNA pattern is well known as the most influential climate patterns in the Northern Hemisphere mid-latitudes beyond the tropics and is strongly influenced by the El Niño-Southern Oscillation phenomenon. However, it was never considered to be influencing the sea level variability in the Baltic Sea (except for some discussed influence on the Baltic Sea ice, Jevrejeva, Moore, and Grinsted 2003). Our analysis showed for the first time that PNA could affect the Baltic Sea extreme sea levels in intermittent periods. This can occur during the transition period of the regime shifts when the relationship with NAO is weak.

1. Sometimes in the text, Fig and Figure are used indistinctly.

- The names of the figures were unified

2. Line 190. Figure 4 is called before figure 3 in the main text.

- In the new version of the manuscript, it is corrected

3. Line 220/225. In order to assure reproducibility, I would recommend indicating the equations, functions, or tools used in Matlab as well as a quick description of Hydrognomon, for those who are not familiar with it.

- Thank you, we added the description in the manuscript

4. In the same sense, I recommend an extension of the explanation about the water level data processing before performing the distribution fitting. By doing so, the reader can reproduce and understand how the data have been processed to obtain the extreme water level from the tide gauge records.

- The additional description was added to the manuscript

5. Figures. The overall quality of the figures should be improved, for instance, by matching the font size and font name with the main text.

- The figures were improved

6. Line 260. I would recall here, as the authors do in section 2.3, that changes in the shape parameter have consequences in the tail of the distribution, determining the behavior of the events with very low frequency.

- We added the additional description of the consequences of the tail distribution

7. Line 270, figure 5. It would be good to see results for all other locations, as in figure 7. Same for the results of scale parameters (figure 6).

- The figures for the other locations are added

8. Line 290. “As the 95% confidence intervals of estimates of this parameter for single years mostly overlap, it is safe to say that the location parameter of the GEV distribution of water level extremes has not experienced any substantial changes in Latvian waters since the 1960s”. I’m curious about changes in the location parameter at Skulte, these changes appear significant when comparing the location parameter obtained by $\sim$2000 vs $\sim$1975. Also, including a grid in the figures would improve the comparison across years. I recommend also to further explain how the statistical significance has been calculated.

- What we meant is that although visually there is a change in the location parameter, the uncertainties in Fig.5 are rather large. Considering the uncertainties, the location

parameter could have been ∼98 (lower level) in 1975 and ∼99 (upper level) in 2001. The uncertainties (95% conf. intervals) were obtained during the GEV fitting. We added a more detailed explanation about it.

9. Line 415. "... observed and measured water maxima in...". Aren't they the same?

- The typo was corrected

Please see below Fig. 1: Sliding correlation results for Daugavgriva and NAO, SCAND, and PNA climatic indices. The correlation is calculated for the water level monthly maxima with a window of 15 years. Only the correlation coefficients with the significance of more than 95% are shown.

References

Bueh, C. and Nakamura, H.: Scandinavian pattern and its climatic impact, Quarterly Journal of the Royal Meteorological Society 133(629), 2117–2131, doi:10.1002/qj.173, 2007.

Gao, T., Yu, J.-Y., and Paek, H.: Impacts of four northern-hemisphere teleconnection patterns on atmospheric circulations over Eurasia and the Pacific, Theoretical and Applied Climatology, 129(3–4), 815–831, doi:10.1007/s00704-016-1801-2, 2017.

Jevrejeva, S., Moore, J.C., and Grinsted, A.: Influence of the Arctic Oscillation and El Niño-Southern Oscillation (ENSO) on ice conditions in the Baltic Sea: The wavelet approach, Journal of Geophysical Research - Atmospheres, 108(D21), 4677, doi: 10.1029/2003JD003417, 2003.

Jevrejeva, S., Moore, J. C., Woodworth, P. L., and Grinsted, A.: Influence of large-scale atmospheric circulation on European sea level: results based on the wavelet transform method, Tellus A, 57, 183–193, 2005.

Karabil, S., Zorita, E., and Hünicke B. 2017. Mechanisms of variability in decadal sea-level trends in the Baltic Sea over the 20th century. Earth System Dynamics, 8,

1031–1046, https://doi.org/10.5194/esd-8-1031-2017, 2017

Karabil, S., Zorita, E., and Hünicke, B.: Contribution of atmospheric circulation to recent off-shore sea-level variations in the Baltic Sea and the North Sea, Earth System Dynamics, 9(1), 69–90, doi: 10.5194/esd-9-69-2018, 2018.

Trigo, R.M., Osborn, T.J., and Corte-Real, J.M.: The North Atlantic Oscillation influence on Europe: climate impacts and associated physical mechanisms, Climate Research 20(1), 9–17. https://doi.org/10.3354/cr020009, 2002.

Please also note the supplement to this comment:
https://nhess.copernicus.org/preprints/nhess-2020-100/nhess-2020-100-AC2-supplement.pdf
* * *
[Figure]

Daugavgrīva
NAO
SCAND
PNA
Month
Time

1.0
0.8
0.6
0.4
0.2
-1.0
-0.8
-0.6
-0.4
-0.2

**Fig. 1.**

---

## Referee Report (RR1)

Review of **Non-stationary analysis of water level extremes in Latvian waters, Baltic Sea, during 1961–2018.**

The authors have improved the previous manuscript by adding a section on climate indices, which partially explains the temporal variability of the GEV parameters. This provides additional value to the initial work, also offering a first approach to understanding the sources of this variation in the study area. However, I still have some concerns related to the criterion of the level of statistical significance, which appears to be arbitrary. The importance of this relies on the fact that one of the main results (the linear trend in the GEV parameters) is based on the definition of this level of statistical significance.

In addition to this, minor comments and erratum are included in this second revision. Also, according to the previous review, I recommend improving the overall quality of the figures.

- **Abstract, lines 19-20.** It states that significant linear trends in location and scale parameters are found in Liepāja and Kolka, however, the linear trend in the scale parameter is not significant at Kolka but in the shape (info in paragraph 409).
- In the same sense, the third highlight is incorrect (**line 30**).
- **Line 111.** In this context "observed" and "measured" mean the same thing since they are tide gauge records. I would avoid redundancies.
- **Line 119.** You mention here that data from Parnü have been analyzed in a previous study. It seems this information is better placed in line 173, where you again speak about previous studies that analyzed data from tide gauges in the area.
- **Line 221.** The same information about particular cases of GEV distribution is mentioned two lines above.
- **Line 254.** The brackets seem to be misplaced. Similar comment in line 291 about quotes.
- **Line 321 (page 14).** You mention that the location parameter doesn't show significant changes, I would include here which significance level is considered.
- **Line 339.** The sentence "*some locations and spatially variable pattern of its variations*" sounds confusing.
- **Paragraph starting in line 409**. If I understood correctly, the linear trends are not statistically significant (>95%) in any of the GEV parameters anywhere. You use the 95%

criteria of statistical significance in section 3.4. However, you lower your significance criterion to 80% here, so the linear trends in two parameters in Liepaja and Kolka becomes statistically significant. This fact allows you to highlight the differences in water level extremes between the sites inside and outside the inner area of the Gulf. Therefore, the 80% statistical significance criterion appear to ne non- objective.

In the same vine, the paragraph ends stating that "*the presented features indicate an intrinsic difference in the behavior of the water levels extremes in the inner area of the Gulf of Riga compared to the stations that reflect water level in the Baltic proper*", which is not correct since the trends are not statistically significant. If the 80% significance criterion is accepted, you might include a table showing the significance level found for all other sites, so that the reader can actually make a comparison between those sites in the inner area and those outside it.

It would also be interesting to see the values of the linear trends when significant. For instance, one might expect the linear trend in the location parameter in Kolka to be very small, but this is not shown in the paper.

- **Line 450.** In Figure 6, it is included the 50-yr return levels for all sites. However, this is not discussed in the main text (neither in methods or in results sections).
- **Line 531.** Add the statistical level you are using.

- **Figures.**

1) You are using "Fig." and "Figure" indistinctly (line 161). I also wonder if you have to use the same nomenclature as in the figure legends (when you always use "Figure").
2) The numbering of the figures is incorrect from Figure 3 onwards. There is no agreement between the legend in the figures and the main text.
3) I strongly encourage you to improve the quality of the figures, including (using the numbering of the figures):
    3.1) the overall quality of the graphics (sometimes they seem blurry, Figures 3 and 4 are an example);
    3.2) matching the font size of the figures with the main text as much as possible and also the font name. Figure 3 is an example of different font sizes that makes it

a bit messy. Also, the labels on the y-axis as well as the legends are so small that are difficult to read.

3.3) Figure 3 can perhaps be improved by trying a combination of horizontal and vertical positions of the subplots so you can make them bigger.

3.4) same layout for figures when possible. For instance, the cases of Figure 3 to Figure 5.

---

## Author Response (AR2)

Reply to the referee 2.

We thank the anonymous referee for careful reading of the manuscript and useful comments. Please find below the detailed reply to each comment.

*- Abstract, lines 19-20. It states that significant linear trends in location and scale parameters are found in Liepāja and Kolka, however, the linear trend in the scale parameter is not significant at Kolka but in the shape (info in paragraph 409).*

The abstract was changed accordingly.

*- In the same sense, the third highlight is incorrect (line 30).*

The highlights were updated.

*- Line 111. In this context "observed" and "measured" mean the same thing since they are tide gauge records. I would avoid redundancies.*

The "observed" was removed from the sentence.

*- Line 119. You mention here that data from Parnü have been analyzed in a previous study. It seems this information is better placed in line 173, where you again speak about previous studies that analyzed data from tide gauges in the area.*

The sentences about the data from Pärnu were moved.

*- Line 221. The same information about particular cases of GEV distribution is mentioned two lines above.*

The repetition was removed.

*- Line 254. The brackets seem to be misplaced. Similar comment in line 291 about quotes.*

The brackets and quotes were corrected.

*- Line 321 (page 14). You mention that the location parameter doesn't show significant changes, I would include here which significance level is considered.*

The significance level was added.

*- Line 339. The sentence "some locations and spatially variable pattern of its variations" sounds confusing.*

The phrase was re-written.

*- Paragraph starting in line 409. If I understood correctly, the linear trends are not statistically significant (>95%) in any of the GEV parameters anywhere. You use the 95% criteria of statistical significance in section 3.4. However, you lower your significance criterion to 80% here, so the linear trends in two parameters in Liepaja and Kolka becomes statistically significant. This fact allows you to highlight the differences in water level extremes between the sites inside and outside*

*the inner area of the Gulf. Therefore, the 80% statistical significance criterion appear to ne non-objective.*

*In the same vine, the paragraph ends stating that "the presented features indicate an intrinsic difference in the behavior of the water levels extremes in the inner area of the Gulf of Riga compared to the stations that reflect water level in the Baltic proper", which is not correct since the trends are not statistically significant. If the 80% significance criterion is accepted, you might include a table showing the significance level found for all other sites, so that the reader can actually make a comparison between those sites in the inner area and those outside it.*

*It would also be interesting to see the values of the linear trends when significant. For instance, one might expect the linear trend in the location parameter in Kolka to be very small, but this is not shown in the paper.*

We rephrased some sentences saying that there is a tentative indication of different behavior of water level extremes in the Gulf of Riga and Baltic Proper. The value of the linear trend in shape parameter at Kolka was also added.

*- Line 450. In Figure 6, it is included the 50-yr return levels for all sites. However, this is not discussed in the main text (neither in methods or in results sections).*

We added a discussion of the shown return levels.

*- Line 531. Add the statistical level you are using.*

The level was added.

*- Figures.*

*1) You are using "Fig." and "Figure" indistinctly (line 161). I also wonder if you have to use the same nomenclature as in the figure legends (when you always use "Figure").*

In line 161 the Figure was changed to Fig. The figure legends have the full word "Figure" according to the journal word template. The abbreviation "Fig." was used when it appears in running text according to the journal rules.

*2) The numbering of the figures is incorrect from Figure 3 onwards. There is no agreement between the legend in the figures and the main text.*

Thank you for noticing this, somehow the numbering of figures got changed when the manuscript was edited in different word versions. The numbering was corrected. All the figure numbers in the text were correct.

*3) I strongly encourage you to improve the quality of the figures, including (using the numbering of the figures):*

*3.1) the overall quality of the graphics (sometimes they seem blurry, Figures 3 and 4 are an example);*

*3.2) matching the font size of the figures with the main text as much as possible and also the font name. Figure 3 is an example of different font sizes that makes it a bit messy. Also, the labels on the y-axis as well as the legends are so small that are difficult to read.*

*3.3) Figure 3 can perhaps be improved by trying a combination of horizontal and vertical positions of the subplots so you can make them bigger.*

*3.4) same layout for figures when possible. For instance, the cases of Figure 3 to Figure 5.*

The quality of figures was improved, especially in Figure 3.